# Scalable Sampling for Nonsymmetric Determinantal Point Processes

**Insu Han**
Yale University
insu.han@yale.edu

**Mike Gartrell**
Criteo AI Lab
m.gartrell@criteo.com

**Jennifer Gillenwater**
Google Research
jengi@google.com

**Elvis Dohmatob**
Facebook AI Research
dohmatob@fb.com

**Amin Karbasi**
Yale University
amin.karbasi@yale.edu

## ABSTRACT

A determinantal point process (DPP) on a collection of $M$ items is a model, parameterized by a symmetric kernel matrix, that assigns a probability to every subset of those items. Recent work shows that removing the kernel symmetry constraint, yielding nonsymmetric DPPs (NDPPs), can lead to significant predictive performance gains for machine learning applications. However, existing work leaves open the question of scalable NDPP sampling. There is only one known DPP sampling algorithm, based on Cholesky decomposition, that can directly apply to NDPPs as well. Unfortunately, its runtime is cubic in $M$, and thus does not scale to large item collections. In this work, we first note that this algorithm can be transformed into a linear-time one for kernels with low-rank structure. Furthermore, we develop a scalable sublinear-time rejection sampling algorithm by constructing a novel proposal distribution. Additionally, we show that imposing certain structural constraints on the NDPP kernel enables us to bound the rejection rate in a way that depends only on the kernel rank. In our experiments we compare the speed of all of these samplers for a variety of real-world tasks.

## 1 INTRODUCTION

A determinantal point process (DPP) on $M$ items is a model, parameterized by a symmetric kernel matrix, that assigns a probability to every subset of those items. DPPs have been applied to a wide range of machine learning tasks, including stochastic gradient descent (SGD) (Zhang et al., 2017), reinforcement learning (Osogami & Raymond, 2019; Yang et al., 2020), text summarization (Dupuy & Bach, 2018), coresets (Tremblay et al., 2019), and more. However, a symmetric kernel can only capture negative correlations between items. Recent works (Brunel, 2018; Gartrell et al., 2019) have shown that using a nonsymmetric DPP (NDPP) allows modeling of positive correlations as well, which can lead to significant predictive performance gains. Gartrell et al. (2021) provides scalable NDPP kernel learning and MAP inference algorithms, but leaves open the question of scalable sampling. The only known sampling algorithm for NDPPs is the Cholesky-based approach described in Poulson (2019), which has a runtime of $O(M^3)$ and thus does not scale to large item collections.

There is a rich body of work on efficient sampling algorithms for (symmetric) DPPs, including recent works such as Derezinski et al. (2019); Poulson (2019); Calandriello et al. (2020). Key distinctions between existing sampling algorithms include whether they are for exact or approximate sampling, whether they assume the DPP kernel has some low-rank $K \ll M$, and whether they sample from the space of all $2^M$ subsets or from the restricted space of size-$k$ subsets, so-called $k$-DPPs. In the context of MAP inference, influential work, including Summa et al. (2014); Chen et al. (2018); Hassani et al. (2019); Ebrahimi et al. (2017); Indyk et al. (2020), proposed efficient algorithms that the approximate (sub)determinant maximization problem and provide rigorous guarantees. In this work we focus on exact sampling for low-rank kernels, and provide scalable algorithms for NDPPs. Our contributions are as follows, with runtime and memory details summarized in Table 1:

• Linear-time sampling (Section 3): We show how to transform the $O(M^3)$ Cholesky-decomposition-based sampler from Poulson (2019) into an $O(MK^2)$ sampler for rank-$K$ kernels.

Table 1: Runtime and memory complexities for sampling algorithms developed in this work. $M$ is the size of the entire item set (ground set), and $K$ is the rank of the kernel (often $K \ll M$ in practice). We use $k$ by the size of the sampled set (often $k \ll K$ in practice). $\omega \in [0, 1]$ is a data-dependent constant (with our specific learning scheme, $\omega \ll 1$). The sublinear-time rejection algorithm includes a one-time preprocessing step, after which each successive sample only requires "sampling time".

| Sampling algorithm | Preprocessing time | Sampling time | Memory |
|---|---|---|---|
| Linear-time Cholesky-based | – | $O(MK^2)$ | $O(MK)$ |
| Sublinear-time rejection | $O(MK^2)$ | $O((k^3 \log M + k^4 + K)(1+\omega)^K)$ * | $O(MK^2)$ |

* This assumes some orthogonality constraint on the kernel.

- Sublinear-time sampling (Section 4): Using rejection sampling, we show how to leverage existing sublinear-time samplers for symmetric DPPs to implement a sublinear-time sampler for a subclass of NDPPs that we call orthogonal NDPPs (ONDPPs).

- Learning with orthogonality constraints (Section 5): We show that the scalable NDPP kernel learning of Gartrell et al. (2021) can be slightly modified to impose an orthogonality constraint, yielding the ONDPP subclass. The constraint allows us to control the rejection sampling algorithm's rejection rate, ensuring its scalability. Experiments suggest that the predictive performance of the kernels is not degraded by this change.

For a common large-scale setting where $M$ is 1 million, our sublinear-time sampler results in runtime that is hundreds of times faster than the linear-time sampler. In the same setting, our linear-time sampler provides runtime that is millions of times faster than the only previously known NDPP sampling algorithm, which has cubic time complexity and is thus impractical in this scenario.

## 2    BACKGROUND

**Notation.** We use $[M] := \{1, \ldots, M\}$ to denote the set of items 1 through $M$. We use $\boldsymbol{I}_K$ to denote the $K$-by-$K$ identity matrix, and often write $\boldsymbol{I} := \boldsymbol{I}_M$ when the dimensionality should be clear from context. Given $\boldsymbol{L} \in \mathbb{R}^{M \times M}$, we use $\boldsymbol{L}_{i,j}$ to denote the entry in the $i$-th row and $j$-th column, and $\boldsymbol{L}_{A,B} \in \mathbb{R}^{|A| \times |B|}$ for the submatrix formed by taking rows $A$ and columns $B$. We also slightly abuse notation to denote principal submatrices with a single subscript, $\boldsymbol{L}_A := \boldsymbol{L}_{A,A}$.

**Kernels.** As discussed earlier, both (symmetric) DPPs and NDPPs define a probability distribution over all $2^M$ subsets of a ground set $[M]$. The distribution is parameterized by a kernel matrix $\boldsymbol{L} \in \mathbb{R}^{M \times M}$ and the probability of a subset $Y \subseteq [M]$ is defined to be $\Pr(Y) \propto \det(\boldsymbol{L}_Y)$. For this to define a valid distribution, it must be the case that $\det(\boldsymbol{L}_Y) \geq 0$ for all $Y$. For symmetric DPPs, the non-negativity requirement is identical to a requirement that $\boldsymbol{L}$ be positive semi-definite (PSD). For nonsymmetric DPPs, there is no such simple correspondence, but prior work such as Gartrell et al. (2019; 2021) has focused on PSD matrices for simplicity.

**Normalizing and marginalizing.** The normalizer of a DPP or NDPP distribution can also be written as a single determinant: $\sum_{Y \subseteq [M]} \det(\boldsymbol{L}_Y) = \det(\boldsymbol{L} + \boldsymbol{I})$ (Kulesza & Taskar, 2012, Theorem 2.1). Additionally, the marginal probability of a subset can be written as a determinant: $\Pr(A \subseteq Y) = \det(\boldsymbol{K}_A)$, for $\boldsymbol{K} := \boldsymbol{I} - (\boldsymbol{L} + \boldsymbol{I})^{-1}$ (Kulesza & Taskar, 2012, Theorem 2.2)*, where $\boldsymbol{K}$ is typically called the marginal kernel.

**Intuition.** The diagonal element $\boldsymbol{K}_{i,i}$ is the probability that item $i$ is included in a set sampled from the model. The 2-by-2 determinant $\det(\boldsymbol{K}_{\{i,j\}}) = \boldsymbol{K}_{i,i}\boldsymbol{K}_{j,j} - \boldsymbol{K}_{i,j}\boldsymbol{K}_{j,j}$ is the probability that both $i$ and $j$ are included in the sample. A symmetric DPP has a symmetric marginal kernel, meaning $\boldsymbol{K}_{i,j} = \boldsymbol{K}_{j,i}$, and hence $\boldsymbol{K}_{i,i}\boldsymbol{K}_{j,j} - \boldsymbol{K}_{i,j}\boldsymbol{K}_{j,i} \leq \boldsymbol{K}_{i,i}\boldsymbol{K}_{j,j}$. This implies that the probability of including both $i$ and $j$ in the sampled set cannot be greater than the product of their individual inclusion probabilities. Hence, symmetric DPPs can only encode negative correlations. In contrast, NDPPs can have $\boldsymbol{K}_{i,j}$ and $\boldsymbol{K}_{j,i}$ with differing signs, allowing them to also capture positive correlations.

### 2.1    RELATED WORK

**Learning.** Gartrell et al. (2021) proposes a low-rank kernel decomposition for NDPPs that admits linear-time learning. The decomposition takes the form $\boldsymbol{L} := \boldsymbol{V}\boldsymbol{V}^\top + \boldsymbol{B}(\boldsymbol{D} - \boldsymbol{D}^\top)\boldsymbol{B}^\top$ for

---

*The proofs in Kulesza & Taskar (2012) typically assume a symmetric kernel, but this particular one does not rely on the symmetry.

---

**Algorithm 1** Cholesky-based NDPP sampling (Poulson, 2019, Algorithm 1)

1: **procedure** SAMPLECHOLESKY($\boldsymbol{K}$)                    ▷ marginal kernel factorization $\boldsymbol{Z}, \boldsymbol{W}$
2:     $Y \leftarrow \emptyset$                                                                              $\boldsymbol{Q} \leftarrow \boldsymbol{W}$
3:     **for** $i = 1$ **to** $M$ **do**
4:         $p_i \leftarrow \boldsymbol{K}_{i,i}$                                                              $p_i \leftarrow \boldsymbol{z}_i^\top \boldsymbol{Q} \boldsymbol{z}_i$
5:         $u \leftarrow \text{uniform}(0, 1)$
6:         **if** $u \leq p_i$ **then** $Y \leftarrow Y \cup \{i\}$
7:         **else** $p_i \leftarrow p_i - 1$
8:         $\boldsymbol{K}_A \leftarrow \boldsymbol{K}_A - \frac{\boldsymbol{K}_{A,i}\boldsymbol{K}_{i,A}}{p_i}$ for $A := \{i+1, \ldots, M\}$         $\boldsymbol{Q} \leftarrow \boldsymbol{Q} - \frac{\boldsymbol{Q}\boldsymbol{z}_i \boldsymbol{z}_i^\top \boldsymbol{Q}}{p_i}$
9:     **return** $Y$

---

$\boldsymbol{V}, \boldsymbol{B} \in \mathbb{R}^{M \times K}$, and $\boldsymbol{D} \in \mathbb{R}^{K \times K}$. The $\boldsymbol{V}\boldsymbol{V}^\top$ component is a rank-$K$ symmetric matrix, which can model negative correlations between items. The $\boldsymbol{B}(\boldsymbol{D} - \boldsymbol{D}^\top)\boldsymbol{B}^\top$ component is a rank-$K$ skew-symmetric matrix, which can model positive correlations between items. For compactness of notation, we will write $\boldsymbol{L} = \boldsymbol{Z}\boldsymbol{X}\boldsymbol{Z}^\top$, where $\boldsymbol{Z} = \begin{bmatrix} \boldsymbol{V} & \boldsymbol{B} \end{bmatrix} \in \mathbb{R}^{M \times 2K}$, and $\boldsymbol{X} = \begin{bmatrix} \boldsymbol{I}_K & \boldsymbol{0} \\ \boldsymbol{0} & \boldsymbol{D} - \boldsymbol{D}^\top \end{bmatrix} \in \mathbb{R}^{2K \times 2K}$. The marginal kernel in this case also has a rank-$2K$ decomposition, as can be shown via application of the Woodbury matrix identity:

$$\boldsymbol{K} := \boldsymbol{I} - (\boldsymbol{I} + \boldsymbol{L})^{-1} = \boldsymbol{Z}\boldsymbol{X}\left(\boldsymbol{I}_{2K} + \boldsymbol{Z}^\top \boldsymbol{Z} \boldsymbol{X}\right)^{-1} \boldsymbol{Z}^\top. \tag{1}$$

Note that the matrix to be inverted can be computed from $\boldsymbol{Z}$ and $\boldsymbol{X}$ in $O(MK^2)$ time, and the inverse itself takes $O(K^3)$ time. Thus, $\boldsymbol{K}$ can be computed from $\boldsymbol{L}$ in time $O(MK^2)$. We will develop sampling algorithms for this decomposition, as well as an orthogonality-constrained version of it. We use $\boldsymbol{W} := \boldsymbol{X}\left(\boldsymbol{I}_{2K} + \boldsymbol{Z}^\top \boldsymbol{Z} \boldsymbol{X}\right)^{-1}$ in what follows so that we can compactly write $\boldsymbol{K} = \boldsymbol{Z}\boldsymbol{W}\boldsymbol{Z}^\top$.

**Sampling.** While there are a number of exact sampling algorithms for DPPs with symmetric kernels, the only published algorithm that clearly can directly apply to NDPPs is from Poulson (2019) (see Theorem 2 therein). This algorithm begins with an empty set $Y = \emptyset$ and iterates through the $M$ items, deciding for each whether or not to include it in $Y$ based on all of the previous inclusion/exclusion decisions. Poulson (2019) shows, via the Cholesky decomposition, that the necessary conditional probabilities can be computed as follows:

$$\Pr\left(j \in Y \mid i \in Y\right) = \frac{\Pr(\{i, j\} \subseteq Y)}{\Pr(i \in Y)} = \boldsymbol{K}_{j,j} - \left(\boldsymbol{K}_{j,i}\boldsymbol{K}_{i,j}\right)/\boldsymbol{K}_{i,i}, \tag{2}$$

$$\Pr\left(j \in Y \mid i \notin Y\right) = \frac{\Pr(j \in Y) - \Pr(\{i, j\} \subseteq Y)}{\Pr(i \notin Y)} = \boldsymbol{K}_{j,j} - \left(\boldsymbol{K}_{j,i}\boldsymbol{K}_{i,j}\right)/\left(\boldsymbol{K}_{i,i} - 1\right). \tag{3}$$

Algorithm 1 (left-hand side) gives pseudocode for this Cholesky-based sampling algorithm[†]. There has also been some recent work on *approximate* sampling for fixed-size $k$-NDPPs: Alimohammadi et al. (2021) provide a Markov chain Monte Carlo (MCMC) algorithm and prove that the overall runtime to approximate $\varepsilon$-close total variation distance is bounded by $O(M^2 k^3 \log(1/(\varepsilon \Pr(Y_0))))$, where $\Pr(Y_0)$ is probability of an initial state $Y_0$. Improving this runtime is an interesting avenue for future work, but for this paper we focus on exact sampling.

## 3  LINEAR-TIME CHOLESKY-BASED SAMPLING

In this section, we show that the $O(M^3)$ runtime of the Cholesky-based sampler from Poulson (2019) can be significantly improved when using the low-rank kernel decomposition of Gartrell et al. (2021). First, note that Line 8 of Algorithm 1, where all marginal probabilities are updated via an $(M - i)$-by-$(M - i)$ matrix subtraction, is the most costly part of the algorithm, making overall time and memory complexities $O(M^3)$ and $O(M^2)$, respectively. However, when the DPP kernel is given by a low-rank decomposition, we observe that marginal probabilities can be updated by matrix-vector

---

[†]Cholesky decomposition is defined only for a symmetric positive definite matrix. However, we use the term "Cholesky" from Poulson (2019) to maintain consistency with this work, although Algorithm 1 is valid for nonsymmetric matrices.

---

**Algorithm 2** Rejection NDPP sampling                                     (Tree-based sampling)

---

1: **procedure** PREPROCESS($\boldsymbol{V}, \boldsymbol{B}, \boldsymbol{D}$)

2:     $\{(\sigma_j, \boldsymbol{y}_{2j-1}, \boldsymbol{y}_{2j})\}_{j=1}^{K/2} \leftarrow$ YOULADECOMPOSE($\boldsymbol{B}, \boldsymbol{D}$)[‡]

3:     $\widehat{\boldsymbol{X}} \leftarrow \mathrm{diag}\big(\boldsymbol{I}_K, \sigma_1, \sigma_1, \ldots, \sigma_{K/2}, \sigma_{K/2}\big)$

4:     $\boldsymbol{Z} \leftarrow [\boldsymbol{V}, \boldsymbol{y}_1, \ldots, \boldsymbol{y}_K]$                 $\{(\lambda_i, \boldsymbol{z}_i)\}_{i=1}^{2K} \leftarrow$ EIGENDECOMPOSE($\boldsymbol{Z}\widehat{\boldsymbol{X}}^{1/2}$)

              $\mathcal{T} \leftarrow$ CONSTRUCTTREE($M, [\boldsymbol{z}_1, \ldots, \boldsymbol{z}_{2K}]^\top$)

5:     **return** $\boldsymbol{Z}, \widehat{\boldsymbol{X}}$                                  **return** $\mathcal{T}, \{(\lambda_i, \boldsymbol{z}_i)\}_{i=1}^{2K}$

---

6: **procedure** SAMPLEREJECT($\boldsymbol{V}, \boldsymbol{B}, \boldsymbol{D}, \boldsymbol{Z}, \hat{\boldsymbol{X}}$)     ▷ tree $\mathcal{T}$, eigen pair $\{(\lambda_i, \boldsymbol{z}_i)\}_{i=1}^{2K}$ of $\boldsymbol{Z}\widehat{\boldsymbol{X}}\boldsymbol{Z}$

7:     **while** true **do**

8:         $Y \leftarrow$ SAMPLEDPP($\boldsymbol{Z}\widehat{\boldsymbol{X}}\boldsymbol{Z}^\top$)                 $Y \leftarrow$ SAMPLEDPP($\mathcal{T}, \{(\lambda_i, \boldsymbol{z}_i)\}_{i=1}^{2K}$)

9:         $u \leftarrow \mathrm{uniform}(0, 1)$

10:        $p \leftarrow \dfrac{\det([\boldsymbol{V}\boldsymbol{V}^\top + \boldsymbol{B}(\boldsymbol{D}-\boldsymbol{D}^\top)\boldsymbol{B}^\top]_Y)}{\det([\boldsymbol{Z}\widehat{\boldsymbol{X}}\boldsymbol{Z}^\top]_Y)}$

11:        **if** $u \leq p$ **then break**

12:    **return** $Y$

---

multiplications of dimension $2K$, regardless of $M$. In more detail, suppose we have the marginal kernel $\boldsymbol{K} = \boldsymbol{Z}\boldsymbol{W}\boldsymbol{Z}^\top$ as in Eq. (1) and let $\boldsymbol{z}_j$ be the $j$-th row vector in $\boldsymbol{Z}$. Then, for $i \neq j$:

$$\Pr(j \in Y \mid i \in Y) = \boldsymbol{K}_{j,j} - (\boldsymbol{K}_{j,i}\boldsymbol{K}_{i,j})/\boldsymbol{K}_{i,i} = \boldsymbol{z}_j^\top \left(\boldsymbol{W} - \frac{(\boldsymbol{W}\boldsymbol{z}_i)(\boldsymbol{z}_i^\top \boldsymbol{W})}{\boldsymbol{z}_i^\top \boldsymbol{W}\boldsymbol{z}_i}\right)\boldsymbol{z}_j, \quad (4)$$

$$\Pr(j \in Y \mid i \notin Y) = \boldsymbol{z}_j^\top \left(\boldsymbol{W} - \frac{(\boldsymbol{W}\boldsymbol{z}_i)(\boldsymbol{z}_i^\top \boldsymbol{W})}{\boldsymbol{z}_i^\top \boldsymbol{W}\boldsymbol{z}_i - 1}\right)\boldsymbol{z}_j. \quad (5)$$

The conditional probabilities in Eqs. (4) and (5) are of bilinear form, and the $\boldsymbol{z}_j$ do not change during sampling. Hence, it is enough to update the $2K$-by-$2K$ inner matrix at each iteration, and obtain the marginal probability by multiplying this matrix by $\boldsymbol{z}_i$. The details are shown on the right-hand side of Algorithm 1. The overall time and memory complexities are $O(MK^2)$ and $O(MK)$, respectively.

## 4 SUBLINEAR-TIME REJECTION SAMPLING

Although the Cholesky-based sampler runs in time linear in $M$, even this is too expensive for the large $M$ that are often encountered in real-world datasets. To improve runtime, we consider *rejection sampling* (Von Neumann, 1963). Let $p$ be the target distribution that we aim to sample, and let $q$ be any distribution whose support corresponds to that of $p$; we call $q$ the proposal distribution. Assume that there is a universal constant $U$ such that $p(x) \leq Uq(x)$ for all $x$. In this setting, rejection sampling draws a sample $x$ from $q$ and accepts it with probability $p(x)/(Uq(x))$, repeating until an acceptance occurs. The distribution of the resulting samples is $p$. It is important to choose a good proposal distribution $q$ so that sampling is efficient and the number of rejections is small.

### 4.1 PROPOSAL DPP CONSTRUCTION

Our first goal is to find a proposal DPP with symmetric kernel $\widehat{\boldsymbol{L}}$ that can upper-bound all probabilities of samples from the NDPP with kernel $\boldsymbol{L}$ within a constant factor. To this end, we expand the determinant of a principal submatrix, $\det(\boldsymbol{L}_Y)$, using the spectral decomposition of the NDPP kernel. Such a decomposition essentially amounts to combining the eigendecomposition of the symmetric part of $\boldsymbol{L}$ with the Youla decomposition (Youla, 1961) of the skew-symmetric part.

Specifically, suppose $\{(\sigma_j, \boldsymbol{y}_{2j-1}, \boldsymbol{y}_{2j})\}_{j=1}^{K/2}$ is the Youla decomposition of $\boldsymbol{B}(\boldsymbol{D} - \boldsymbol{D}^\top)\boldsymbol{B}^\top$ (see Appendix D for more details), that is,

$$\boldsymbol{B}(\boldsymbol{D} - \boldsymbol{D}^\top)\boldsymbol{B}^\top = \sum_{j=1}^{K/2} \sigma_j \left(\boldsymbol{y}_{2j-1}\boldsymbol{y}_{2j}^\top - \boldsymbol{y}_{2j}\boldsymbol{y}_{2j-1}^\top\right). \quad (6)$$

---

[‡]Pseudo-code of YOULADECOMPOSE is provided in Algorithm 4. See Appendix D.

Then we can simply write $\boldsymbol{L} = \boldsymbol{Z}\boldsymbol{X}\boldsymbol{Z}^\top$, for $\boldsymbol{Z} := [\boldsymbol{V}, \boldsymbol{y}_1, \ldots, \boldsymbol{y}_K] \in \mathbb{R}^{M \times 2K}$, and

$$\boldsymbol{X} := \mathrm{diag}\left(\boldsymbol{I}_K, \begin{bmatrix} 0 & \sigma_1 \\ -\sigma_1 & 0 \end{bmatrix}, \ldots, \begin{bmatrix} 0 & \sigma_{K/2} \\ -\sigma_{K/2} & 0 \end{bmatrix}\right). \tag{7}$$

Now, consider defining a related but *symmetric* PSD kernel $\widehat{\boldsymbol{L}} := \boldsymbol{Z}\widehat{\boldsymbol{X}}\boldsymbol{Z}^\top$ with $\widehat{\boldsymbol{X}} := \mathrm{diag}\left(\boldsymbol{I}_K, \sigma_1, \sigma_1, \ldots, \sigma_{K/2}, \sigma_{K/2}\right)$. All determinants of the principal submatrices of $\widehat{\boldsymbol{L}} = \boldsymbol{Z}\widehat{\boldsymbol{X}}\boldsymbol{Z}^\top$ upper-bound those of $\boldsymbol{L}$, as stated below.

**Theorem 1.** *For every subset $Y \subseteq [M]$, it holds that $\det(\boldsymbol{L}_Y) \leq \det(\widehat{\boldsymbol{L}}_Y)$. Moreover, equality holds when the size of $Y$ is equal to the rank of $\boldsymbol{L}$.*

*Proof sketch*: From the Cauchy-Binet formula, the determinants of $\boldsymbol{L}_Y$ and $\widehat{\boldsymbol{L}}_Y$ for all $Y \subseteq [M], |Y| \leq 2K$ can be represented as

$$\det(\boldsymbol{L}_Y) = \sum_{I \subseteq [K], |I|=|Y|} \sum_{J \subseteq [K], |J|=|Y|} \det(\boldsymbol{X}_{I,J}) \det(\boldsymbol{Z}_{Y,I}) \det(\boldsymbol{Z}_{Y,J}), \tag{8}$$

$$\det(\widehat{\boldsymbol{L}}_Y) = \sum_{I \subseteq [2K], |I|=|Y|} \det(\widehat{\boldsymbol{X}}_I) \det(\boldsymbol{Z}_{Y,I})^2. \tag{9}$$

Many of the terms in Eq. (8) are actually zero due to the block-diagonal structure of $\boldsymbol{X}$. For example, note that if $1 \in I$ but $1 \notin J$, then there is an all-zeros row in $\boldsymbol{X}_{I,J}$, making $\det(\boldsymbol{X}_{I,J}) = 0$. We show that each $\boldsymbol{X}_{I,J}$ with nonzero determinant is a block-diagonal matrix with diagonal entries among $\pm\sigma_j$, or $\begin{bmatrix} 0 & \sigma_j \\ -\sigma_j & 0 \end{bmatrix}$. With this observation, we can prove that $\det(\boldsymbol{X}_{I,J})$ is upper-bounded by $\det(\widehat{\boldsymbol{X}}_I)$ or $\det(\widehat{\boldsymbol{X}}_J)$. Then, through application of the rearrangement inequality, we can upper-bound the sum of the $\det(\boldsymbol{X}_{I,J}) \det(\boldsymbol{Z}_{Y,I}) \det(\boldsymbol{Z}_{Y,J})$ in Eq. (8) with a sum over $\det(\widehat{\boldsymbol{X}}_I) \det(\boldsymbol{Z}_{Y,I})^2$. Finally, we show that the number of non-zero terms in Eq. (8) is identical to the number of non-zero terms in Eq. (9). Combining these gives us the desired inequality $\det(\boldsymbol{L}_Y) \leq \det(\widehat{\boldsymbol{L}}_Y)$. The full proof of Theorem 1 is in Appendix E.1.

Now, recall that the normalizer of a DPP (or NDPP) with kernel $\boldsymbol{L}$ is $\det(\boldsymbol{L} + \boldsymbol{I})$. The ratio of probability of the NDPP with kernel $\boldsymbol{L}$ to that of a DPP with kernel $\widehat{\boldsymbol{L}}$ is thus:

$$\frac{\mathrm{Pr}_{\boldsymbol{L}}(Y)}{\mathrm{Pr}_{\widehat{\boldsymbol{L}}}(Y)} = \frac{\det(\boldsymbol{L}_Y)/\det(\boldsymbol{L}+\boldsymbol{I})}{\det(\widehat{\boldsymbol{L}}_Y)/\det(\widehat{\boldsymbol{L}}+\boldsymbol{I})} \leq \frac{\det(\widehat{\boldsymbol{L}}+\boldsymbol{I})}{\det(\boldsymbol{L}+\boldsymbol{I})},$$

where the inequality follows from Theorem 1. This gives us the necessary universal constant $U$ upper-bounding the ratio of the target distribution to the proposal distribution. Hence, given a sample $Y$ drawn from the DPP with kernel $\widehat{\boldsymbol{L}}$, we can use acceptance probability $\mathrm{Pr}_{\boldsymbol{L}}(Y)/(U \mathrm{Pr}_{\widehat{\boldsymbol{L}}}(Y)) = \det(\boldsymbol{L}_Y)/\det(\widehat{\boldsymbol{L}}_Y)$. Pseudo-codes for proposal construction and rejection sampling are given in Algorithm 2. Note that to derive $\widehat{\boldsymbol{L}}$ from $\boldsymbol{L}$ it suffices to run the Youla decomposition of $\boldsymbol{B}(\boldsymbol{D} - \boldsymbol{D}^\top)\boldsymbol{B}^\top$, because the difference is only in the skew-symmetric part. This decomposition can run in $O(MK^2)$ time; more details are provided in Appendix D. Since $\widehat{\boldsymbol{L}}$ is a symmetric PSD matrix, we can apply existing fast DPP sampling algorithms to sample from it. In particular, in the next section we combine a fast tree-based method with rejection sampling.

### 4.2 SUBLINEAR-TIME TREE-BASED SAMPLING

There are several DPP sampling algorithms that run in sublinear time, such as tree-based (Gillenwater et al., 2019) and intermediate (Derezinski et al., 2019) sampling algorithms. Here, we consider applying the former, a tree-based approach, to sample from the proposal distribution defined by $\widehat{\boldsymbol{L}}$.

We give some details of the sampling procedure, as in the course of applying it we discovered an optimization that slightly improves on the runtime of prior work. Formally, let $\{(\lambda_i, \boldsymbol{z}_i)\}_{i=1}^{2K}$ be the eigendecomposition of $\widehat{\boldsymbol{L}}$ and $\boldsymbol{Z} := [\boldsymbol{z}_1, \ldots, \boldsymbol{z}_{2K}] \in \mathbb{R}^{M \times 2K}$. As shown in Kulesza & Taskar (2012, Lemma 2.6), for every $Y \subseteq [M], |Y| \leq 2K$, the probability of $Y$ under DPP with $\widehat{\boldsymbol{L}}$ can be written:

$$\mathrm{Pr}_{\widehat{\boldsymbol{L}}}(Y) = \frac{\det(\widehat{\boldsymbol{L}}_Y)}{\det(\widehat{\boldsymbol{L}}+\boldsymbol{I})} = \sum_{E \subseteq [2K], |E|=|Y|} \det(\boldsymbol{Z}_{Y,E}\boldsymbol{Z}_{Y,E}^\top) \prod_{i \in E} \frac{\lambda_i}{\lambda_i + 1} \prod_{i \notin E} \frac{1}{\lambda_i + 1}. \tag{10}$$

---

**Algorithm 3** Tree-based DPP sampling (Gillenwater et al., 2019)

1: **procedure** BRANCH($A, \boldsymbol{Z}$)
2:    **if** $A = \{j\}$ **then**
3:       $\mathcal{T}.A \leftarrow \{j\}, \mathcal{T}.\boldsymbol{\Sigma} \leftarrow \boldsymbol{Z}_{j,:}^{\top}\boldsymbol{Z}_{j,:}$
4:       **return** $\mathcal{T}$
5:    $A_\ell, A_r \leftarrow$ Split $A$ in half
6:    $\mathcal{T}.\texttt{left} \leftarrow$ BRANCH($A_\ell, \boldsymbol{Z}$)
7:    $\mathcal{T}.\texttt{right} \leftarrow$ BRANCH($A_r, \boldsymbol{Z}$)
8:    $\mathcal{T}.\boldsymbol{\Sigma} \leftarrow \mathcal{T}.\texttt{left}.\boldsymbol{\Sigma} + \mathcal{T}.\texttt{right}.\boldsymbol{\Sigma}$
9:    **return** $\mathcal{T}$

10: **procedure** CONSTRUCTTREE($M, \boldsymbol{Z}$)
11:    **return** BRANCH($[M], \boldsymbol{Z}$)

---

12: **procedure** SAMPLEDPP($\mathcal{T}, \boldsymbol{Z}, \{\lambda_i\}_{i=1}^{K}$)
13:    $E \leftarrow \emptyset, Y \leftarrow \emptyset, \boldsymbol{Q}^Y \leftarrow \boldsymbol{0}$
14:    **for** $i = 1, \ldots, K$ **do**
15:       $E \leftarrow E \cup \{i\}$ w.p. $\lambda_i/(\lambda_i + 1)$
16:    **for** $k = 1, \ldots, |E|$ **do**
17:       $j \leftarrow$ SAMPLEITEM($\mathcal{T}, \boldsymbol{Q}^Y, E$)
18:       $Y \leftarrow Y \cup \{j\}$
19:       $\boldsymbol{Q}^Y \leftarrow \boldsymbol{I}_{|E|} - \boldsymbol{Z}_{Y,E}^{\top}\left(\boldsymbol{Z}_{Y,E}\boldsymbol{Z}_{Y,E}^{\top}\right)^{-1}\boldsymbol{Z}_{Y,E}$
20:    **return** $Y$

21: **procedure** SAMPLEITEM($\mathcal{T}, \boldsymbol{Q}^Y, E$)
22:    **if** $\mathcal{T}$ is a leaf **then return** $\mathcal{T}.A$
23:    $p_\ell \leftarrow \langle \mathcal{T}.\texttt{left}.\boldsymbol{\Sigma}_E, \boldsymbol{Q}^Y \rangle$
24:    $p_r \leftarrow \langle \mathcal{T}.\texttt{right}.\boldsymbol{\Sigma}_E, \boldsymbol{Q}^Y \rangle$
25:    $u \leftarrow$ uniform$(0, 1)$
26:    **if** $u \leq \frac{p_\ell}{p_\ell + p_r}$ **then**
27:       **return** SAMPLEITEM($\mathcal{T}.\texttt{left}, \boldsymbol{Q}^Y, E$)
28:    **else**
29:       **return** SAMPLEITEM($\mathcal{T}.\texttt{right}, \boldsymbol{Q}^Y, E$)

---

A matrix of the form $\boldsymbol{Z}_{:,E}\boldsymbol{Z}_{:,E}^{\top}$ can be a valid marginal kernel for a special type of DPP, called an elementary DPP. Hence, Eq. (10) can be thought of as DPP probabilities expressed as a mixture of elementary DPPs. Based on this mixture view, DPP sampling can be done in two steps: (1) choose an elementary DPP according to its mixture weight, and then (2) sample a subset from the selected elementary DPP. Step (1) can be performed by $2K$ independent random coin tossings, while step (2) involves computational overhead. The key idea of tree-based sampling is that step (2) can be accelerated by traversing a binary tree structure, which can be done in time logarithmic in $M$.

More specifically, given the marginal kernel $\boldsymbol{K} = \boldsymbol{Z}_{:,E}\boldsymbol{Z}_{:,E}^{\top}$, where $E$ is obtained from step (1), we start from the empty set $Y = \emptyset$ and repeatedly add an item $j$ to $Y$ with probability:

$$\Pr(j \in S \mid Y \subseteq S) = \boldsymbol{K}_{j,j} - \boldsymbol{K}_{j,Y}(\boldsymbol{K}_Y)^{-1}\boldsymbol{K}_{Y,j} = \boldsymbol{Z}_{j,E}\boldsymbol{Q}^Y\boldsymbol{Z}_{j,E}^{\top} = \langle \boldsymbol{Q}^Y, (\boldsymbol{Z}_{j,:}^{\top}\boldsymbol{Z}_{j,:})_E \rangle, \quad (11)$$

where $S$ is some final selected subset, and $\boldsymbol{Q}^Y := \boldsymbol{I}_{|E|} - \boldsymbol{Z}_{Y,E}^{\top}\left(\boldsymbol{Z}_{Y,E}\boldsymbol{Z}_{Y,E}^{\top}\right)^{-1}\boldsymbol{Z}_{Y,E}$. Consider a binary tree whose root includes a ground set $[M]$. Every non-leaf node contains a subset $A \subseteq [M]$ and stores a $2K$-by-$2K$ matrix $\sum_{j \in A} \boldsymbol{Z}_{j,:}^{\top}\boldsymbol{Z}_{j,:}$. A partition $A_\ell$ and $A_r$, such that $A_\ell \cup A_r = A$, $A_\ell \cap A_r = \emptyset$, are passed to its left and right subtree, respectively. The resulting tree has $M$ leaves and each has exactly a single item. Then, one can sample a single item by recursively moving down to the left node with probability:

$$p_\ell = \frac{\langle \boldsymbol{Q}^Y, \sum_{j \in A_\ell}(\boldsymbol{Z}_{j,:}^{\top}\boldsymbol{Z}_{j,:})_E \rangle}{\langle \boldsymbol{Q}^Y, \sum_{j \in A}(\boldsymbol{Z}_{j,:}^{\top}\boldsymbol{Z}_{j,:})_E \rangle}, \tag{12}$$

or to the right node with probability $1 - p_\ell$, until reaching a leaf node. An item in the leaf node is chosen with probability according to Eq. (11). Since every subset in the support of an elementary DPP with a rank-$k$ kernel has exactly $k$ items, this process is repeated for $|E|$ iterations. Full descriptions of tree construction and sampling are provided in Algorithm 3. The proposed tree-based rejection sampling for an NDPP is outlined on the right-side of Algorithm 2. The one-time pre-processing step of constructing the tree (CONSTRUCTTREE) requires $O(MK^2)$ time. After pre-processing, the procedure SAMPLEDPP involves $|E|$ traversals of a tree of depth $O(\log M)$, where in each node a $O(|E|^2)$ operation is required. The overall runtime is summarized in Proposition 1 and the proof can be found in Appendix E.2.

**Proposition 1.** *The tree-based sampling procedure* SAMPLEDPP *in Algorithm 3 runs in time* $O(K + k^3 \log M + k^4)$, *where $k$ is the size of the sampled set*[§].

---

[§]Computing $p_\ell$ via Eq. (12) improves on Gillenwater et al. (2019)'s $O(k^4 \log M)$ runtime for this step.

### 4.3 AVERAGE NUMBER OF REJECTIONS

We now return to rejection sampling and focus on the expected number of rejections. The number of rejections of Algorithm 2 is known to be a geometric random variable with mean equal to the constant $U$ used to upper-bound the ratio of the target distribution to the proposal distribution: $\det(\widehat{L} + I)/\det(L + I)$. If all columns in $V$ and $B$ are orthogonal, which we denote $V \perp B$, then the expected number of rejections depends only on the eigenvalues of the skew-symmetric part of the NDPP kernel.

**Theorem 2.** *Given an NDPP kernel* $L = VV^\top + B(D - D^\top)B^\top$ *for* $V, B \in \mathbb{R}^{M \times K}, D \in \mathbb{R}^{K \times K}$, *consider the proposal kernel* $\widehat{L}$ *as proposed in Section 4.1. Let* $\{\sigma_j\}_{j=1}^{K/2}$ *be the positive eigenvalues obtained from the Youla decomposition of* $B(D - D^\top)B^\top$. *If* $V \perp B$, *then* $\frac{\det(\widehat{L}+I)}{\det(L+I)} = \prod_{j=1}^{K/2} \left(1 + \frac{2\sigma_j}{\sigma_j^2+1}\right) \leq (1+\omega)^{K/2}$, *where* $\omega = \frac{2}{K} \sum_{j=1}^{K/2} \frac{2\sigma_j}{\sigma_j^2+1} \in (0, 1]$.

*Proof sketch*: Orthogonality between $V$ and $B$ allows $\det(L + I)$ to be expressed just in terms of the eigenvalues of $VV^\top$ and $B(D - D^\top)B^\top$. Since both $L$ and $\widehat{L}$ share the symmetric part $VV^\top$, the ratio of determinants only depends on the skew-symmetric part. A more formal proof appears in Appendix E.3.

Assuming we have a kernel where $V \perp B$, we can combine Theorem 2 with the tree-based rejection sampling algorithm (right-side in Algorithm 2) to sample in time $O((K + k^3 \log M + k^4)(1+\omega)^{K/2})$. Hence, we have a sampling algorithm that is sublinear in $M$, and can be much faster than the Cholesky-based algorithm when $(1 + \omega)^{K/2} \ll M$. In the next section, we introduce a learning scheme with the $V \perp B$ constraint, as well as regularization to ensure that $\omega$ is small.

## 5 LEARNING WITH ORTHOGONALITY CONSTRAINTS

We aim to learn a NDPP that provides both good predictive performance and a low rejection rate. We parameterize our NDPP kernel matrix $L = VV^\top + B(D - D^\top)B^\top$ by

$$D = \text{diag}\left(\begin{bmatrix} 0 & \sigma_1 \\ 0 & 0 \end{bmatrix}, \ldots, \begin{bmatrix} 0 & \sigma_{K/2} \\ 0 & 0 \end{bmatrix}\right) \tag{13}$$

for $\sigma_j \geq 0$, $B^\top B = I$, and, motivated by Theorem 2, require $V^\top B = 0$[¶]. We call such orthogonality-constrained NDPPs "ONDPPs". Notice that if $V \perp B$, then $L$ has the full rank of $2K$, since the intersection of the column spaces spanned by $V$ and by $B$ is empty, and thus the full rank available for modeling can be used. Thus, this constraint can also be thought of as simply ensuring that ONDPPs use the full rank available to them.

Given example subsets $\{Y_1, \ldots, Y_n\}$ as training data, learning is done by minimizing the regularized negative log-likelihood:

$$\min_{V, B, \{\sigma_j\}_{j=1}^{K/2}} -\frac{1}{n} \sum_{i=1}^{n} \log\left(\frac{\det(L_{Y_i})}{\det(L+I)}\right) + \alpha \sum_{i=1}^{M} \frac{\|v_i\|_2^2}{\mu_i} + \beta \sum_{i=1}^{M} \frac{\|b_i\|_2^2}{\mu_i} + \gamma \sum_{j=1}^{K/2} \log\left(1 + \frac{2\sigma_j}{\sigma_j^2+1}\right) \tag{14}$$

where $\alpha, \beta, \gamma > 0$ are hyperparameters, $\mu_i$ is the frequency of item $i$ in the training data, and $v_i$ and $b_i$ represent the rows of $V$ and $B$, respectively. This objective is very similar to that of Gartrell et al. (2021), except for the orthogonality constraint and the final regularization term. Note that this regularization term corresponds exactly to the logarithm of the average rejection rate, and therefore should help to control the number of rejections.

## 6 EXPERIMENTS

We first show that the orthogonality constraint from Section 5 does not degrade the predictive performance of learned kernels. We then compare the speed of our proposed sampling algorithms.

---

[¶]Technical details: To learn NDPP models with the constraint $V^\top B = 0$, we project $V$ according to: $V \leftarrow V - B(B^\top B)^{-1}(B^\top V)$. For the $B^\top B = I$ constraint, we apply QR decomposition on $B$. Note that both operations require $O(MK^2)$ time. (Constrained learning and sampling code is provided at https://github.com/insuhan/nonsymmetric-dpp-sampling. We use Pytorch's linalg.solve to avoid the expense of explicitly computing the $(B^\top B)^{-1}$ inverse.) Hence, our learning time complexity is identical to that of Gartrell et al. (2021).

Table 2: Average MPR and AUC, with 95% confidence estimates obtained via bootstrapping, test log-likelihood, and the number of rejections for NDPP models. Bold values indicate the best MPR, outside of the confidence intervals of the two baseline methods.

| Low-rank DPP Models | Metric | UK Retail $M$=3,941 | Recipe $M$=7,993 | Instacart $M$=49,677 | Million Song $M$=371,410 | Book $M$=1,059,437 |
|---|---|---|---|---|---|---|
| Symmetric DPP (Gartrell et al., 2017) | MPR | 76.42 ± 0.97 | 95.04 ± 0.69 | 93.06 ± 0.92 | 90.00 ± 1.18 | 72.54 ± 2.03 |
| | AUC | 0.74 ± 0.01 | 0.99 ± 0.01 | 0.86 ± 0.01 | 0.77 ± 0.01 | 0.70 ± 0.01 |
| | Log-Likelihood | -104.89 | -44.63 | -73.22 | -310.14 | -149.76 |
| NDPP (Gartrell et al., 2021) | MPR | 77.09 ± 1.10 | 95.17 ± 0.67 | 92.40 ± 1.05 | 89.00 ± 1.11 | 72.98 ± 1.46 |
| | AUC | 0.74 ± 0.01 | 0.99 ± 0.00 | 0.87 ± 0.01 | 0.80 ± 0.01 | 0.74 ± 0.01 |
| | Log-Likelihood | -99.09 | -44.72 | -74.94 | -314.12 | -149.93 |
| | # of Rejections | $4.136 \times 10^{10}$ | 78.95 | $6.806 \times 10^3$ | $3.907 \times 10^{10}$ | $9.245 \times 10^6$ |
| ONDPP without regularization | MPR | **78.43 ± 0.95** | 95.40 ± 0.62 | 92.80 ± 0.99 | **93.02 ± 0.83** | 75.35 ± 1.83 |
| | AUC | 0.71 ± 0.00 | 0.99 ± 0.01 | 0.83 ± 0.01 | 0.77 ± 0.01 | 0.64 ± 0.01 |
| | Log-Likelihood | -99.45 | -44.60 | -72.69 | -302.64 | -140.53 |
| | # of Rejections | $1.818 \times 10^9$ | 103.81 | 128.96 | $5.563 \times 10^7$ | 682.22 |
| ONDPP with regularization | MPR | 77.12 ± 0.98 | 95.50 ± 0.59 | 92.99 ± 0.95 | 92.86 ± 0.80 | **75.73 ± 1.84** |
| | AUC | 0.72 ± 0.01 | 0.99 ± 0.01 | 0.83 ± 0.01 | 0.77 ± 0.01 | 0.64 ± 0.01 |
| | Log-Likelihood | -103.83 | -44.56 | -72.72 | -305.66 | -140.67 |
| | # of Rejections | 26.09 | 21.59 | 79.74 | 45.42 | 61.10 |

## 6.1 PREDICTIVE PERFORMANCE RESULTS FOR NDPP LEARNING

We benchmark various DPP models, including symmetric (Gartrell et al., 2017), nonsymmetric for scalable learning (Gartrell et al., 2021), as well as our ONDPP kernels with and without rejection rate regularization. We use the scalable NDPP models (Gartrell et al., 2021) as a baseline[||]. The kernel components of each model are learned using five real-world recommendation datasets, which have ground set sizes that range from 3,941 to 1,059,437 items (see Appendix A for more details).

Our experimental setup and metrics mirror those of Gartrell et al. (2021). We report the mean percentile rank (MPR) metric for a next-item prediction task, the AUC metric for subset discrimination, and the log-likelihood of the test set; see Appendix B for more details on the experiments and metrics. For all metrics, higher numbers are better. For NDPP models, we additionally report the average rejection rates when they apply to rejection sampling.

In Table 2, we observe that the predictive performance of our ONDPP models generally match or sometimes exceed the baseline. This is likely because the orthogonality constraint enables more effective use of the full rank-$2K$ feature space. Moreover, imposing the regularization on rejection rate, as shown in Eq. (14), often leads to dramatically smaller rejection rates, while the impact on predictive performance is generally marginal. These results justify the ONDPP and regularization for fast sampling. Finally, we observe that the learning time of our ONDPP models is typically a bit longer than that of the NDPP models, but still quite reasonable (e.g., the time per iteration for the NDPP takes 27 seconds for the Book dataset, while our ONDPP takes 49.7 seconds).

Fig. 1 shows how the regularizer $\gamma$ affects the test log-likelihood and the average number of rejections. We see that $\gamma$ degrades predictive performance and reduces the rejection rate when set above a certain threshold; this behavior is seen for many datasets. However, for the Recipe dataset we observed that the test log-likelihood is not very sensitive to $\gamma$, likely because all models in our experiments achieve very high performance on this dataset. In general, we observe that $\gamma$ can be set to a value that results in a small rejection rate, while having minimal impact on predictive performance.

## 6.2 SAMPLING TIME COMPARISON

We benchmark the Cholesky-based sampling algorithm (Algorithm 1) and tree-based rejection sampling algorithm (Algorithm 2) on ONDPPs with both synthetic and real-world data.

---

[||]We use the code from https://github.com/cgartrel/scalable-nonsymmetric-DPPs for the NDPP baseline, which is made available under the MIT license. To simplify learning and MAP inference, Gartrell et al. (2021) set $B = V$ in their experiments. However, since we have the $V \perp B$ constraint in our ONDPP approach, we cannot set $B = V$. Hence, for a fair comparison, we do not set $B = V$ for the NDPP baseline in our experiments, and thus the results in Table 2 differ slightly from those published in Gartrell et al. (2021).

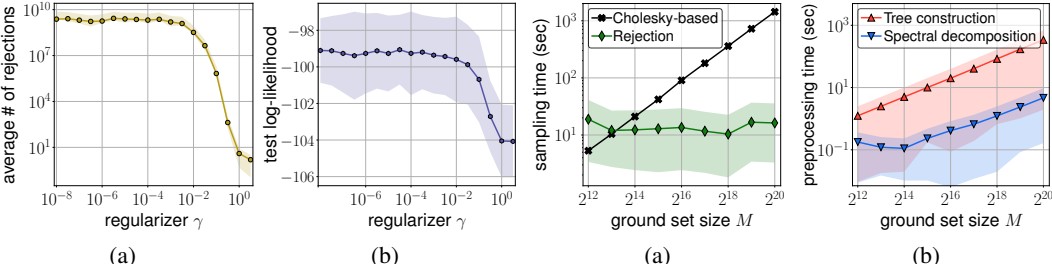

Figure 1: Average number of rejections and test log-likelihood with different values of the regularizer $\gamma$ for ONDPPs trained on the UK Retail dataset. Shaded regions are 95% confidence intervals of 10 independent trials.

Figure 2: Wall-clock time (sec) for synthetic data for (a) NDPP sampling algorithms and (b) preprocessing steps for the rejection sampling. Shaded regions are 95% confidence intervals from 100 independent trials.

Table 3: Wall-clock time (sec) for preprocessing and sampling ONDPPs trained on real-world data, and speedup of the tree-based sampler over the Cholesky-based one. We set $K = 100$ and provide average times with 95% confidence intervals from 10 independent trials for the Cholesky-based algorithm and 100 trials for the rejection algorithm. Memory usage for the tree is also reported.

| | UK Retail $M$=3,941 | Recipe $M$=7,993 | Instacart $M$=49,677 | Million Song $M$=371,410 | Book $M$=1,059,437 |
|---|---|---|---|---|---|
| Spectral decomposition | 0.209 | 0.226 | 0.505 | 2.639 | 7.482 |
| Tree construction | 0.997 | 1.998 | 12.65 | 119.0 | 340.1 |
| Cholesky-based sampling | $5.572 \pm 0.056$ | $11.36 \pm 0.098$ | $71.82 \pm 1.087$ | $545.8 \pm 8.776$ | $1{,}631 \pm 11.84$ |
| Tree-based rejection sampling | $2.463 \pm 0.417$ | $1.331 \pm 0.241$ | $5.962 \pm 1.049$ | $14.72 \pm 2.620$ | $6.627 \pm 1.294$ |
| (Speedup) | ($\times 2.262$) | ($\times 8.535$) | ($\times 12.05$) | ($\times 37.08$) | ($\times 246.1$) |
| Tree memory usage | 630.5 MB | 1.279 GB | 7.948 GB | 59.43 GB | 169.5 GB |

**Synthetic datasets.** We generate non-uniform random features for $V, B$ as done by (Han & Gillenwater, 2020). In particular, we first sample $x_1, \ldots, x_{100}$ from $\mathcal{N}(0, I_{2K}/(2K))$, and integers $t_1, \ldots, t_{100}$ from Poisson distribution with mean 5, rescaling the integers such that $\sum_i t_i = M$. Next, we draw $t_i$ random vectors from $\mathcal{N}(x_i, I_{2K})$, and assign the first $K$-dimensional vectors as the row vectors of $V$ and the latter vectors as those of $B$. Each entry of $D$ is sampled from $\mathcal{N}(0, 1)$. We choose $K = 100$ and vary $M$ from $2^{12}$ to $2^{20}$.

Fig. 2(a) illustrates the runtimes of Algorithms 1 and 2. We verify that the rejection sampling time tends to increase sub-linearly with the ground set size $M$, while the Cholesky-based sampler runs in linear time. In Fig. 2(b), the runtimes of the preprocessing steps for Algorithm 2 (i.e., spectral decomposition and tree construction) are reported. Although the rejection sampler requires these additional processes, they are one-time steps and run much faster than a single run of the Choleksy-based method for $M = 2^{20}$.

**Real-world datasets.** In Table 3, we report the runtimes and speedup of NDPP sampling algorithms for real-world datasets. All NDPP kernels are obtained using learning with orthogonality constraints, with rejection rate regularization as reported in Section 6.1. We observe that the tree-based rejection sampling runs up to 246 times faster than the Cholesky-based algorithm. For larger datasets, we expect that this gap would significantly increase. As with the synthetic experiments, we see that the tree construction pre-processing time is comparable to the time required to draw a single sample via the other methods, and thus the tree-based method is often the best choice for repeated sampling[**].

## 7 CONCLUSION

In this work we developed scalable sampling methods for NDPPs. One limitation of our rejection sampler is its practical restriction to the ONDPP subclass. Other opportunities for future work include the extension of our rejection sampling approach to the generation of fixed-size samples (from $k$-NDPPs), the development of approximate sampling techniques, and the extension of DPP samplers along the lines of Derezinski et al. (2019); Calandriello et al. (2020) to NDPPs. Scalable sampling also opens the door to using NDPPs as building blocks in probabilistic models.

---

[**]We note that the tree can consume substantial memory, e.g., 169.5 GB for the Book dataset with $K = 100$. For settings where this scale of memory use is unacceptable, we suggest use of the intermediate sampling algorithm (Calandriello et al., 2020) in place of tree-based sampling. The resulting sampling algorithm may be slower, but the $O(M + K)$ memory cost is substantially lower.

## 8 ETHICS STATEMENT

In general, our work moves in a positive direction by substantially decreasing the computational costs of NDPP sampling. When using our constrained learning method to learn kernels from user data, we recommend employing a technique such as differentially-private SGD (Abadi et al., 2016) to help prevent user data leaks, and adjusting the weights on training examples to balance the impact of sub-groups of users so as to make the final kernel as fair as possible. As far as we are aware, the datasets used in this work do not contain personally identifiable information or offensive content. We were not able to determine if user consent was explicitly obtained by the organizations that constructed these datasets.

## 9 REPRODUCIBILITY STATEMENT

We have made extensive effort to ensure that all algorithmic, theoretical, and experimental contributions described in this work are reproducible. All of the code implementing our constrained learning and sampling algorithms is publicly available [††]. The proofs for our theoretical contributions are available in Appendix E. For our experiments, all dataset processing steps, experimental procedures, and hyperparameter settings are described in Appendices A, B, and C, respectively.

## 10 ACKNOWLEDGEMENTS

Amin Karbasi acknowledges funding in direct support of this work from NSF (IIS-1845032) and ONR (N00014-19-1-2406).

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

## A    FULL DETAILS ON DATASETS

We perform experiments on several real-world public datasets composed of subsets:

• **UK Retail:** This dataset (Chen et al., 2012) contains baskets representing transactions from an online retail company that sells all-occasion gifts. We omit baskets with more than 100 items, leaving us with a dataset containing 19,762 baskets drawn from a catalog of $M = 3,941$ products. Baskets containing more than 100 items are in the long tail of the basket-size distribution, so omitting these is reasonable, and allows us to use a low-rank factorization of the NDPP with $K = 100$.

• **Recipe:** This dataset (Majumder et al., 2019) contains recipes and food reviews from Food.com (formerly Genius Kitchen)[‡‡]. Each recipe ("basket") is composed of a collection of ingredients, resulting in 178,265 recipes and a catalog of 7,993 ingredients.

• **Instacart:** This dataset (Instacart, 2017) contains baskets purchased by Instacart users[§§]. We omit baskets with more than 100 items, resulting in 3.2 million baskets and a catalog of 49,677 products.

• **Million Song:** This dataset (McFee et al., 2012) contains playlists ("baskets") of songs from Echo Nest users[¶¶]. We trim playlists with more than 100 items, leaving 968,674 playlists and a catalog of 371,410 songs.

• **Book:** This dataset (Wan & McAuley, 2018) contains reviews from the Goodreads book review website, including a variety of attributes describing the items[***]. For each user we build a subset ("basket") containing the books reviewed by that user. We trim subsets with more than 100 books, resulting in 430,563 subsets and a catalog of 1,059,437 books.

As far as we are aware, these datasets do not contain personally identifiable information or offensive content. While the UK Retail dataset is publicly available, we were unable to find a license for it. Also, we were not able to determine if user consent was explicitly obtained by the organizations that constructed these datasets.

## B    FULL DETAILS ON EXPERIMENTAL SETUP AND METRICS

We use 300 randomly-selected baskets as a held-out validation set, for tracking convergence during training and for tuning hyperparameters. Another 2000 random baskets are used for testing, and the rest are used for training. Convergence is reached during training when the relative change in validation log-likelihood is below a predetermined threshold. We use PyTorch with Adam (Kingma & Ba, 2015) for optimization. We initialize $D$ from the standard Gaussian distribution $\mathcal{N}(0, 1)$, while $V$ and $B$ are initialized from the uniform$(0, 1)$ distribution.

**Subset expansion task.** We use greedy conditioning to do next-item prediction (Gartrell et al., 2021, Section 4.2). We compare methods using a standard recommender system metric: mean percentile rank (MPR) (Hu et al., 2008; Li et al., 2010). MPR of 50 is equivalent to random selection; MPR of 100 means that the model perfectly predicts the next item. See Appendix B.1 for a complete description of the MPR metric.

**Subset discrimination task.** We also test the ability of a model to discriminate observed subsets from randomly generated ones. For each subset in the test set, we generate a subset of the same length by drawing items uniformly at random (and we ensure that the same item is not drawn more than once for a subset). We compute the AUC for the model on these observed and random subsets, where the score for each subset is the log-likelihood that the model assigns to the subset.

---

[‡‡]See https://www.kaggle.com/shuyangli94/food-com-recipes-and-user-interactions for the license for this public dataset.

[§§]This public dataset is available for non-commercial use; see https://www.instacart.com/datasets/grocery-shopping-2017 for the license.

[¶¶]See http://millionsongdataset.com/faq/ for the license for this public dataset.

[***]This public dataset is available for academic use only; see https://sites.google.com/eng.ucsd.edu/ucsdbookgraph/home for the license.

### B.1 MEAN PERCENTILE RANK

We begin our definition of MPR by defining percentile rank (PR). First, given a set $J$, let $p_{i,J} = \Pr(J \cup \{i\} \mid J)$. The percentile rank of an item $i$ given a set $J$ is defined as

$$\text{PR}_{i,J} = \frac{\sum_{i' \notin J} \mathbb{1}(p_{i,J} \geq p_{i',J})}{|\mathcal{Y} \backslash J|} \times 100\%$$

where $\mathcal{Y} \backslash J$ indicates those elements in the ground set $\mathcal{Y}$ that are not found in $J$.

For our evaluation, given a test set $Y$, we select a random element $i \in Y$ and compute $\text{PR}_{i,Y \backslash \{i\}}$. We then average over the set of all test instances $\mathcal{T}$ to compute the mean percentile rank (MPR):

$$\text{MPR} = \frac{1}{|\mathcal{T}|} \sum_{Y \in \mathcal{T}} \text{PR}_{i,Y \backslash \{i\}}.$$

## C HYPERPARAMETERS FOR EXPERIMENTS

**Preventing numerical instabilities**: The $\det(\boldsymbol{L}_{Y_i})$ in Eq. (14) will be zero whenever $|Y_i| > K$, where $Y_i$ is an observed subset. To address this in practice we set $K$ to the size of the largest subset observed in the data, $K'$, as in Gartrell et al. (2017). However, this does not entirely fix the issue, as there is still a chance that the term will be zero even when $|Y_i| \leq K$. In this case though, we know that we are not at a maximum, since the value of the objective function is $-\infty$. Numerically, to prevent such singularities, in our implementation we add a small $\epsilon \boldsymbol{I}$ correction to each $\boldsymbol{L}_{Y_i}$ when optimizing Eq. (14) ($\epsilon = 10^{-5}$ in our experiments).

We perform a grid search using a held-out validation set to select the best-performing hyperparameters for each model and dataset. The hyperparameter settings used for each model and dataset are described below.

**Symmetric low-rank DPP** (Gartrell et al., 2017). For this model, we use $K$ for the number of item feature dimensions for the symmetric component $\boldsymbol{V}$, and $\alpha$ for the regularization hyperparameter for $\boldsymbol{V}$. We use the following hyperparameter settings:

- UK Retail dataset: $K = 100, \alpha = 1$.
- Recipe dataset: $K = 100, \alpha = 0.01$
- Instacart dataset: $K = 100, \alpha = 0.001$.
- Million Song dataset: $K = 100, \alpha = 0.0001$.
- Book dataset: $K = 100, \alpha = 0.001$

**Scalable NDPP** (Gartrell et al., 2021). As described in Section 2.1, we use $K$ to denote the number of item feature dimensions for the symmetric component $\boldsymbol{V}$ and the dimensionality of the nonsymmetric component $\boldsymbol{D}$. $\alpha$ and $\beta$ are the regularization hyperparameters. We use the following hyperparameter settings:

- UK dataset: $K = 100, \alpha = 0.01$.
- Recipe dataset: $K = 100, \alpha = \beta = 0.01$.
- Instacart dataset: $K = 100, \alpha = 0.001$.
- Million Song dataset: $K = 100, \alpha = 0.01$.
- Book dataset: $K = 100, \alpha = \beta = 0.1$

**ONDPP**. As described in Section 5, we use $K$ to denote the number of item feature dimensions for the symmetric component $\boldsymbol{V}$ and the dimensionality of the nonsymmetric component $\boldsymbol{C}$. $\alpha$, $\beta$, and $\gamma$ are the regularization hyperparameters. We use the following hyperparameter settings:

- UK dataset: $K = 100, \alpha = \beta = 0.01, \gamma = 0.5$.
- Recipe dataset: $K = 100, \alpha = \beta = 0.01, \gamma = 0.1$.
- Instacart dataset: $K = 100, \alpha = \beta = 0.001, \gamma = 0.001$.
- Million Song dataset: $K = 100, \alpha = \beta = 0.01, \gamma = 0.2$.

- Book dataset: $K = 100, \alpha = \beta = 0.01, \gamma = 0.1$.

For all of the above model configurations and datasets, we use a batch size of 800 during training.

## D  YOULA DECOMPOSITION: SPECTRAL DECOMPOSITION FOR SKEW-SYMMETRIC MATRIX

We provide some basic facts on the spectral decomposition of a skew-symmetric matrix, and introduce an efficient algorithm for this decomposition when it is given by a low-rank factorization. We write $\mathrm{i} := \sqrt{-1}$ and $\boldsymbol{v}^H$ as the conjugate transpose of $\boldsymbol{v} \in \mathbb{C}^M$, and denote $\mathrm{Re}(z)$ and $\mathrm{Im}(z)$ by the real and imaginary parts of a complex number $z$, respectively.

Given $\boldsymbol{B} \in \mathbb{R}^{M \times K}$ and $\boldsymbol{D} \in \mathbb{R}^{K \times K}$, consider a rank-$K$ skew-symmetric matrix $\boldsymbol{B}(\boldsymbol{D} - \boldsymbol{D}^\top)\boldsymbol{B}^\top$. Note that all nonzero eigenvalues of a real-valued skew-symmetric matrix are purely imaginary. Denote $\mathrm{i}\sigma_1, -\mathrm{i}\sigma_1, \ldots, \mathrm{i}\sigma_{K/2}, -\mathrm{i}\sigma_{K/2}$ by its nonzero eigenvalues where each of $\sigma_j$ is real, and $\boldsymbol{a}_1 + \mathrm{i}\boldsymbol{b}_1, \boldsymbol{a}_1 - \mathrm{i}\boldsymbol{b}_1, \ldots \boldsymbol{a}_{K/2} + \mathrm{i}\boldsymbol{b}_{K/2}, \boldsymbol{a}_{K/2} - \mathrm{i}\boldsymbol{b}_{K/2}$ by the corresponding eigenvectors for $\boldsymbol{a}_j, \boldsymbol{b}_j \in \mathbb{R}^M$, which come in conjugate pairs. Then, we can write

$$\boldsymbol{B}(\boldsymbol{D} - \boldsymbol{D}^\top)\boldsymbol{B}^\top = \sum_{j=1}^{K/2} \mathrm{i}\sigma_j(\boldsymbol{a}_j + \mathrm{i}\boldsymbol{b}_j)(\boldsymbol{a}_j + \mathrm{i}\boldsymbol{b}_j)^H - \mathrm{i}\sigma_j(\boldsymbol{a}_j - \mathrm{i}\boldsymbol{b}_j)(\boldsymbol{a}_j - \mathrm{i}\boldsymbol{b}_j)^H \qquad (15)$$

$$= \sum_{j=1}^{K/2} 2\sigma_j(\boldsymbol{a}_j\boldsymbol{b}_j^\top - \boldsymbol{b}_j\boldsymbol{a}_j^\top) \qquad (16)$$

$$= \sum_{j=1}^{K/2} \begin{bmatrix} \boldsymbol{a}_j - \boldsymbol{b}_j & \boldsymbol{a}_j + \boldsymbol{b}_j \end{bmatrix} \begin{bmatrix} 0 & \sigma_j \\ -\sigma_j & 0 \end{bmatrix} \begin{bmatrix} \boldsymbol{a}_j^\top - \boldsymbol{b}_j^\top \\ \boldsymbol{a}_j^\top + \boldsymbol{b}_j^\top \end{bmatrix}. \qquad (17)$$

Note that $\boldsymbol{a}_1 \pm \boldsymbol{b}_1, \ldots, \boldsymbol{a}_{K/2} \pm \boldsymbol{b}_{K/2}$ are real-valued orthonormal vectors, because $\boldsymbol{a}_1, \boldsymbol{b}_1, \ldots, \boldsymbol{a}_{K/2}, \boldsymbol{b}_{K/2}$ are orthogonal to each other and $\|\boldsymbol{a}_j \pm \boldsymbol{b}_j\|_2^2 = \|\boldsymbol{a}_j\|_2^2 + \|\boldsymbol{b}_j\|_2^2 = 1$ for all $j$. The pair $\{(\sigma_j, \boldsymbol{a}_j - \boldsymbol{b}_j, \boldsymbol{a}_j + \boldsymbol{b}_j)\}_{j=1}^{K/2}$ is often called the Youla decomposition (Youla, 1961) of $\boldsymbol{B}(\boldsymbol{D} - \boldsymbol{D}^\top)\boldsymbol{B}^\top$. To efficiently compute the Youla decomposition of a rank-$K$ matrix, we use the following result.

**Proposition 2** (Proposition 1, Nakatsukasa (2019)). *Given $\boldsymbol{A}, \boldsymbol{B} \in \mathbb{C}^{M \times K}$, the nonzero eigenvalues of $\boldsymbol{A}\boldsymbol{B}^\top \in \mathbb{C}^{M \times M}$ and $\boldsymbol{B}^\top\boldsymbol{A} \in \mathbb{C}^{K \times K}$ are identical. In addition, if $(\lambda, \boldsymbol{v})$ is an eigenpair of $\boldsymbol{B}^\top\boldsymbol{A}$ with $\lambda \neq 0$, then $(\lambda, \boldsymbol{A}\boldsymbol{v}/\|\boldsymbol{A}\boldsymbol{v}\|_2)$ is an eigenpair of $\boldsymbol{A}\boldsymbol{B}^\top$.*

From the above proposition, one can first compute $(\boldsymbol{D} - \boldsymbol{D}^\top)\boldsymbol{B}^\top\boldsymbol{B}$ and then apply the eigendecomposition to that $K$-by-$K$ matrix. Taking the imaginary part of the obtained eigenvalues gives us the $\sigma_j$'s, and multiplying $\boldsymbol{B}$ by the eigenvectors gives us the eigenvectors of $\boldsymbol{B}(\boldsymbol{D} - \boldsymbol{D}^\top)\boldsymbol{B}^\top$. In addition, this can be done in $O(MK^2 + K^3)$ time; when $M > K$ it runs much faster than the eigendecomposition of $\boldsymbol{B}(\boldsymbol{D} - \boldsymbol{D}^\top)\boldsymbol{B}^\top$, which requires $O(M^3)$ time. The pseudo-code of the Youla decomposition is provided in Algorithm 4.

---

**Algorithm 4** Youla decomposition of low-rank skew-symmetric matrix

1: **procedure** YOULADECOMPOSITION($\boldsymbol{B}, \boldsymbol{D}$)
2:    $\{(\eta_j, \boldsymbol{z}_j), (\overline{\eta}_j, \overline{\boldsymbol{z}}_j)\}_{j=1}^{K/2} \leftarrow$ eigendecomposition of $(\boldsymbol{D} - \boldsymbol{D}^\top)\boldsymbol{B}^\top\boldsymbol{B}$
3:    **for** $j = 1, \ldots, K/2$ **do**
4:       $\sigma_j \leftarrow \mathrm{Im}(\eta_j)$ for $j = 1, \ldots, K/2$
5:       $\boldsymbol{y}_{2j-1} \leftarrow \boldsymbol{B}\left(\mathrm{Re}(\boldsymbol{z}_j) - \mathrm{Im}(\boldsymbol{z}_j)\right)$
6:       $\boldsymbol{y}_{2j} \leftarrow \boldsymbol{B}\left(\mathrm{Re}(\boldsymbol{z}_j) + \mathrm{Im}(\boldsymbol{z}_j)\right)$
7:    $\boldsymbol{y}_j \leftarrow \boldsymbol{y}_j/\|\boldsymbol{y}_j\|$ for $j = 1, \ldots, K$
8:    **return** $\{(\sigma_j, \boldsymbol{y}_{2j-1}, \boldsymbol{y}_{2j})\}_{j=1}^{K/2}$

---

# E    PROOFS

## E.1    PROOF OF THEOREM 1

**Theorem 1.** *For every subset $Y \subseteq [M]$, it holds that $\det(\boldsymbol{L}_Y) \leq \det(\widehat{\boldsymbol{L}}_Y)$. Moreover, equality holds when the size of $Y$ is equal to the rank of $\boldsymbol{L}$.*

*Proof of Theorem 1.* It is enough to fix $Y \subseteq [M]$ such that $1 \leq |Y| \leq 2K$, because the rank of both $\boldsymbol{L}$ and $\widehat{\boldsymbol{L}}$ is up to $2K$. Denote $k := |Y|$ and $\binom{[2K]}{k} := \{I \subseteq [2K]; |I| = k\}$ for $k \leq 2K$. We recall the definition of $\widehat{\boldsymbol{L}}$: given $\boldsymbol{V}, \boldsymbol{B}, \boldsymbol{D}$ such that $\boldsymbol{L} = \boldsymbol{V}\boldsymbol{V}^\top + \boldsymbol{B}(\boldsymbol{D} - \boldsymbol{D}^\top)\boldsymbol{B}^\top$, let $\{(\rho_i, \boldsymbol{v}_i)\}_{i=1}^K$ be the eigendecomposition of $\boldsymbol{V}\boldsymbol{V}^\top$ and $\{(\sigma_j, \boldsymbol{y}_{2j-1}, \boldsymbol{y}_{2j})\}_{j=1}^{K/2}$ be the Youla decomposition of $\boldsymbol{B}(\boldsymbol{D} - \boldsymbol{D}^\top)\boldsymbol{B}^\top$. Denote $\boldsymbol{Z} := [\boldsymbol{v}_1, \ldots, \boldsymbol{v}_K, \boldsymbol{y}_1, \ldots, \boldsymbol{y}_K] \in \mathbb{R}^{M \times 2K}$ and

$$\boldsymbol{X} := \mathrm{diag}\left(\rho, \ldots, \rho_K, \begin{bmatrix} 0 & \sigma_1 \\ -\sigma_1 & 0 \end{bmatrix}, \ldots, \begin{bmatrix} 0 & \sigma_{K/2} \\ -\sigma_{K/2} & 0 \end{bmatrix}\right),$$

$$\widehat{\boldsymbol{X}} := \mathrm{diag}\left(\rho_1, \ldots, \rho_K, \begin{bmatrix} \sigma_1 & 0 \\ 0 & \sigma_1 \end{bmatrix}, \ldots, \begin{bmatrix} \sigma_{K/2} & 0 \\ 0 & \sigma_{K/2} \end{bmatrix}\right),$$

so that $\boldsymbol{L} = \boldsymbol{Z}\boldsymbol{X}\boldsymbol{Z}^\top$ and $\widehat{\boldsymbol{L}} = \boldsymbol{Z}\widehat{\boldsymbol{X}}\boldsymbol{Z}^\top$. Applying the Cauchy-Binet formula twice, we can write the determinant of the principal submatrices of both $\boldsymbol{L}$ and $\widehat{\boldsymbol{L}}$:

$$\det(\boldsymbol{L}_Y) = \sum_{I \in \binom{[2K]}{k}} \sum_{J \in \binom{[2K]}{k}} \det(\boldsymbol{X}_{I,J}) \det(\boldsymbol{Z}_{Y,I}) \det(\boldsymbol{Z}_{Y,J}), \tag{18}$$

$$\det(\widehat{\boldsymbol{L}}_Y) = \sum_{I \in \binom{[2K]}{k}} \sum_{J \in \binom{[2K]}{k}} \det(\widehat{\boldsymbol{X}}_{I,J}) \det(\boldsymbol{Z}_{Y,I}) \det(\boldsymbol{Z}_{Y,J})$$

$$= \sum_{I \in \binom{[2K]}{k}} \det(\widehat{\boldsymbol{X}}_I) \det(\boldsymbol{Z}_{Y,I})^2, \tag{19}$$

where Eq. (19) follows from the fact that $\widehat{\boldsymbol{X}}$ is diagonal, which means that $\det(\widehat{\boldsymbol{X}}_{I,J}) = 0$ for $I \neq J$.

When the size of $Y$ is equal to the rank of $\boldsymbol{L}$ (i.e., $k = 2K$), the summations in Eqs. (18) and (19) simplify to single terms: $\det(\boldsymbol{L}_Y) = \det(\boldsymbol{X}) \det(\boldsymbol{Z}_{Y,:})^2$ and $\det(\widehat{\boldsymbol{L}}_Y) = \det(\widehat{\boldsymbol{X}}) \det(\boldsymbol{Z}_{Y,:})^2$. Now, observe that the determinants of the full $\boldsymbol{X}$ and $\widehat{\boldsymbol{X}}$ matrices are identical: $\det(\boldsymbol{X}) = \det(\widehat{\boldsymbol{X}}) = \prod_{i=1}^K \rho_i \prod_{j=1}^{K/2} \sigma_j^2$. Hence, it holds that $\det(\boldsymbol{L}_Y) = \det(\widehat{\boldsymbol{L}}_Y)$. This proves the second statement of the theorem.

To prove that $\det(\boldsymbol{L}_Y) \leq \det(\widehat{\boldsymbol{L}}_Y)$ for smaller subsets $Y$, we will use the following:

**Claim 1.** *For every $I, J \in \binom{[2K]}{k}$ such that $\det(\boldsymbol{X}_{I,J}) \neq 0$, there exists a (nonempty) collection of subset pairs $\mathcal{S}(I,J) \subseteq \binom{[2K]}{k} \times \binom{[2K]}{k}$ such that*

$$\sum_{(I',J') \in \mathcal{S}(I,J)} \det(\boldsymbol{X}_{I,J}) \det(\boldsymbol{Z}_{Y,I}) \det(\boldsymbol{Z}_{Y,J}) \leq \sum_{(I',J') \in \mathcal{S}(I,J)} \det(\widehat{\boldsymbol{X}}_{I,I}) \det(\boldsymbol{Z}_{Y,I})^2. \tag{20}$$

**Claim 2.** *The number of nonzero terms in Eq. (18) is identical to that in Eq. (19).*

Combining Claim 1 with Claim 2 yields

$$\det(\boldsymbol{L}_Y) = \sum_{I,J \in \binom{[2K]}{k}} \det(\boldsymbol{X}_{I,J}) \det(\boldsymbol{Z}_{Y,I}) \det(\boldsymbol{Z}_{Y,J}) \leq \sum_{I \in \binom{[2K]}{k}} \det(\widehat{\boldsymbol{X}}_{I,I}) \det(\boldsymbol{Z}_{Y,I})^2 = \det(\widehat{\boldsymbol{L}}_Y).$$

We conclude the proof of Theorem 1. Below we provide proofs for Claim 1 and Claim 2.

*Proof of Claim 1.* Recall that $\boldsymbol{X}$ is a block-diagonal matrix, where each block is of size either 1-by-1, containing $\rho_i$, or 2-by-2, containing both $\sigma_j$ and $-\sigma_j$ in the form $\begin{bmatrix} 0 & \sigma_j \\ -\sigma_j & 0 \end{bmatrix}$. A submatrix $\boldsymbol{X}_{I,J} \in \mathbb{R}^{k \times k}$ with rows $I$ and columns $J$ will only have a nonzero determinant if it contains no

all-zero row or column. Hence, any $\boldsymbol{X}_{I,J}$ with nonzero determinant will have the following form (or some permutation of this block-diagonal):

$$\boldsymbol{X}_{I,J} = \begin{bmatrix} \rho_{p_1} & \cdots & 0 & & & & & & & & \\ \vdots & \ddots & \vdots & & & & & & & \mathbf{0} & \\ 0 & \cdots & \rho_{p_{|P^{I,J}|}} & & & & & & & & \\ & & & \pm\sigma_{q_1} & \cdots & 0 & & & & & \\ & & & \vdots & \ddots & \vdots & & & & & \\ & & & 0 & \cdots & \pm\sigma_{q_{|Q^{I,J}|}} & & & & & \\ & & & & & & 0 & \sigma_{r_1} & & & \\ & & & & & & -\sigma_{r_1} & 0 & & & \\ & & & & & & & & \ddots & & \\ & \mathbf{0} & & & & & & & & 0 & \sigma_{r_{|R^{I,J}|}} \\ & & & & & & & & & -\sigma_{r_{|R^{I,J}|}} & 0 \end{bmatrix} \tag{21}$$

and we denote $P^{I,J} := \{p_1, \ldots, p_{|P^{I,J}|}\}$, $Q^{I,J} := \{q_1, \ldots, q_{|Q^{I,J}|}\}$, and $R^{I,J} := \{r_1, \ldots, r_{|R^{I,J}|}\}$. Indices $p \in P^{I,J}$ yield a diagonal matrix with entries $\rho_p$. For such $p$, both $I$ and $J$ must contain index $p$. Indices $r \in R^{I,J}$ yield a block-diagonal matrix of the form $\begin{bmatrix} 0 & \sigma_r \\ -\sigma_r & 0 \end{bmatrix}$. For such $r$, both $I$ and $J$ must contain a pair of indices, $(K + 2r - 1, K + 2r)$. Finally, indices $q \in Q^{I,J}$ yield a diagonal matrix with entries of $\pm\sigma_q$ (the sign can be $+$ or $-$). For such $q$, $I$ contains $K + 2q - 1$ or $K + 2q$, and $J$ must contain the other. Note that there is no intersection between $Q^{I,J}$ and $R^{I,J}$.

If $Q^{I,J}$ is an empty set (i.e., $I = J$), then $\det(\boldsymbol{X}_{I,J}) = \det(\widehat{\boldsymbol{X}}_{I,J})$ and

$$\det(\boldsymbol{X}_{I,J}) \det(\boldsymbol{Z}_{Y,I}) \det(\boldsymbol{Z}_{Y,J}) = \det(\widehat{\boldsymbol{X}}_I) \det(\boldsymbol{Z}_{Y,I})^2. \tag{22}$$

Thus, the terms in Eq. (18) in this case appear in Eq. (19). Now assume that $Q^{I,J} \neq \emptyset$ and consider the following set of pairs:

$$\mathcal{S}(I, J) := \{(I', J') \; : \; P^{I,J} = P^{I',J'}, Q^{I,J} = Q^{I',J'}, R^{I,J} = R^{I',J'}\}.$$

In other words, for $(I', J') \in \mathcal{S}(I, J)$, the diagonal $\boldsymbol{X}_{I',J'}$ contains $\rho_p$, $\begin{bmatrix} 0 & \sigma_r \\ -\sigma_r & 0 \end{bmatrix}$ exactly as in $\boldsymbol{X}_{I,J}$. However, the signs of the $\sigma_r$'s may differ from $\boldsymbol{X}_{I,J}$. Combining this observation with the definition of $\widehat{\boldsymbol{X}}$,

$$|\det(\boldsymbol{X}_{I',J'})| = |\det(\boldsymbol{X}_{I,J})| = \det(\widehat{\boldsymbol{X}}_I) = \det(\widehat{\boldsymbol{X}}_{I'}) = \det(\widehat{\boldsymbol{X}}_J) = \det(\widehat{\boldsymbol{X}}_{J'}). \tag{23}$$

Therefore,

$$\sum_{(I',J')\in\mathcal{S}(I,J)} \det(\boldsymbol{X}_{I',J'}) \det(\boldsymbol{Z}_{Y,I'}) \det(\boldsymbol{Z}_{Y,J'}) \tag{24}$$

$$\leq \sum_{(I',J')\in\mathcal{S}(I,J)} |\det(\boldsymbol{X}_{I',J'})| \det(\boldsymbol{Z}_{Y,I'}) \det(\boldsymbol{Z}_{Y,J'}) \tag{25}$$

$$= \det(\widehat{\boldsymbol{X}}_I) \sum_{(I',J')\in\mathcal{S}(I,J)} \det(\boldsymbol{Z}_{Y,I'}) \det(\boldsymbol{Z}_{Y,J'}) \tag{26}$$

$$\leq \det(\widehat{\boldsymbol{X}}_I) \sum_{(I',*)\in\mathcal{S}(I,J)} \det(\boldsymbol{Z}_{Y,I'})^2 \tag{27}$$

$$= \sum_{(I',*)\in\mathcal{S}(I,J)} \det(\widehat{\boldsymbol{X}}_{I'}) \det(\boldsymbol{Z}_{Y,I'})^2 \tag{28}$$

where the third line comes from Eq. (23) and the fourth line follows from the rearrangement inequality. Note that application of this inequality does not change the number of terms in the sum. This completes the proof of Claim 1.

*Proof of Claim 2.* In Eq. (19), observe that $\det(\widehat{\boldsymbol{X}}_I)\det(\boldsymbol{Z}_{Y,I})^2 \neq 0$ if and only if $\det(\widehat{\boldsymbol{X}}_I) \neq 0$. Since all $\rho_i$'s and $\sigma_j$'s are positive, the number of $I \subseteq [2K], |I| = k$ such that $\det(\widehat{\boldsymbol{X}}_I) \neq 0$ is equal to $\binom{2K}{k}$. Similarly, the number of nonzero terms in Eq. (18) equals the number of possible choices of $I, J \in \binom{[2K]}{k}$ such that $\det(\boldsymbol{X}_{I,J}) \neq 0$. This can be counted as follows: first choose $i$ items in $\{\rho_1, \ldots, \rho_K\}$ for $i = 0, \ldots, k$; then, choose $j$ items in $\left\{ \left[\begin{smallmatrix} 0 & \sigma_1 \\ -\sigma_1 & 0 \end{smallmatrix}\right], \ldots, \left[\begin{smallmatrix} 0 & \sigma_{K/2} \\ -\sigma_{K/2} & 0 \end{smallmatrix}\right] \right\}$ for $j = 0, \ldots, \lfloor \frac{k-i}{2} \rfloor$; lastly, choose $k - i - 2j$ of $\{\pm\sigma_q; q \notin R^{I,J}\}$, then choose the sign for each of these ($\sigma_q$ or $-\sigma_q$). Combining all of these choices, the total number of nonzero terms is:

$$\sum_{i=0}^{k} \underbrace{\binom{K}{i}}_{\text{choice of } \rho_p} \sum_{j=0}^{\lfloor \frac{k-i}{2} \rfloor} \underbrace{\binom{K/2}{j}}_{\text{choice of } \left[\begin{smallmatrix} 0 & \sigma_r \\ -\sigma_r & 0 \end{smallmatrix}\right]} \underbrace{\binom{K/2 - j}{k - i - 2j} 2^{k-i-2j}}_{\text{choice of } \pm\sigma_q} \tag{29}$$

$$= \sum_{i=0}^{k} \binom{K}{i} \binom{K}{k - i} \tag{30}$$

$$= \binom{2K}{k} \tag{31}$$

where the second line comes from the fact that $\binom{2n}{m} = \sum_{j=0}^{\lfloor \frac{m}{2} \rfloor} \binom{n}{j}\binom{n-j}{m-2j} 2^{m-2j}$ for any integers $n, m \in \mathbb{N}$ such that $m \le 2n$ (see (1.69) in Quaintance (2010)), and the third line follows from the fact that $\sum_{i=0}^{r} \binom{m}{i}\binom{n}{r-i} = \binom{n+m}{r}$ for $n, m, r \in \mathbb{N}$ (Vandermonde's identity). Hence, both the number of nonzero terms in Eqs. (18) and (19) is equal to $\binom{2K}{k}$. This completes the proof of Claim 2. $\qquad\square$

### E.2 PROOF OF PROPOSITION 1

**Proposition 1.** *The tree-based sampling procedure* SAMPLEDPP *in Algorithm 3 runs in time* $O(K + k^3 \log M + k^4)$, *where $k$ is the size of the sampled set*[†††].

*Proof of Proposition 1.* Since computing $p_\ell$ takes $O(k^2)$ from Eq. (12), and since the binary tree has depth $O(\log M)$, SAMPLEITEM in Algorithm 3 runs in $O(k^2 \log M)$ time. Moreover, the query matrix $\boldsymbol{Q}^Y$ can be updated in $O(k^3)$ time as it only requires a $k$-by-$k$ matrix inversion. Therefore, the overall runtime of the tree-based elementary DPP sampling algorithm (after pre-processing) is $O(k^3 \log M + k^4)$. This improves the previous $O(k^4 \log M)$ runtime studied in Gillenwater et al. (2019). Combining this with elementary DPP selection (Line 15 in Algorithm 3), we can sample a set in $O(K + k^3 \log M + k^4)$ time. This completes the proof of Proposition 1. $\qquad\square$

### E.3 PROOF OF THEOREM 2

**Theorem 2.** *Given an NDPP kernel* $\boldsymbol{L} = \boldsymbol{V}\boldsymbol{V}^\top + \boldsymbol{B}(\boldsymbol{D} - \boldsymbol{D}^\top)\boldsymbol{B}^\top$ *for* $\boldsymbol{V}, \boldsymbol{B} \in \mathbb{R}^{M \times K}, \boldsymbol{D} \in \mathbb{R}^{K \times K}$, *consider the proposal kernel* $\widehat{\boldsymbol{L}}$ *as proposed in Section 4.1. Let* $\{\sigma_j\}_{j=1}^{K/2}$ *be the positive eigenvalues obtained from the Youla decomposition of* $\boldsymbol{B}(\boldsymbol{D} - \boldsymbol{D}^\top)\boldsymbol{B}^\top$. *If* $\boldsymbol{V} \perp \boldsymbol{B}$, *then* $\frac{\det(\widehat{\boldsymbol{L}}+\boldsymbol{I})}{\det(\boldsymbol{L}+\boldsymbol{I})} = \prod_{j=1}^{K/2} \left(1 + \frac{2\sigma_j}{\sigma_j^2 + 1}\right) \le (1 + \omega)^{K/2}$, *where* $\omega = \frac{2}{K}\sum_{j=1}^{K/2} \frac{2\sigma_j}{\sigma_j^2 + 1} \in (0, 1]$.

*Proof of Theorem 2.* Since the column spaces of $\boldsymbol{V}$ and $\boldsymbol{B}$ are orthogonal, the corresponding eigenvectors are also orthogonal, i.e., $\boldsymbol{Z}^\top \boldsymbol{Z} = \boldsymbol{I}_{2K}$. Then,

$$\det(\boldsymbol{L} + \boldsymbol{I}) = \det(\boldsymbol{Z}\boldsymbol{X}\boldsymbol{Z}^\top + \boldsymbol{I}) = \det(\boldsymbol{X}\boldsymbol{Z}^\top\boldsymbol{Z} + \boldsymbol{I}_{2K}) = \det(\boldsymbol{X} + \boldsymbol{I}_{2K}) \tag{32}$$

$$= \prod_{i=1}^{K}(\rho_i + 1) \prod_{j=1}^{K/2} \det\left(\begin{bmatrix} 1 & \sigma_j \\ -\sigma_j & 1 \end{bmatrix}\right) \tag{33}$$

$$= \prod_{i=1}^{K}(\rho_i + 1) \prod_{j=1}^{K/2}(\sigma_j^2 + 1) \tag{34}$$

---

[†††]Computing $p_\ell$ via Eq. (12) improves on Gillenwater et al. (2019)'s $O(k^4 \log M)$ runtime for this step.

and similarly

$$\det(\widehat{\boldsymbol{L}} + \boldsymbol{I}) = \prod_{i=1}^{K}(\rho_i + 1)\prod_{j=1}^{K/2}(\sigma_j + 1)^2. \tag{35}$$

Combining Eqs. (34) and (35), we have that

$$\frac{\det(\widehat{\boldsymbol{L}} + \boldsymbol{I})}{\det(\boldsymbol{L} + \boldsymbol{I})} = \prod_{j=1}^{K/2}\frac{(\sigma_j + 1)^2}{(\sigma_j^2 + 1)} = \prod_{j=1}^{K/2}\left(1 + \frac{2\sigma_j}{\sigma_j^2 + 1}\right) \le \left(1 + \frac{2}{K}\sum_{j=1}^{K/2}\frac{2\sigma_j}{\sigma_j^2 + 1}\right)^{K/2} \tag{36}$$

where the inequality holds from the Jensen's inequality. This completes the proof of Theorem 2. $\quad\square$

