# OpenReview forum: "Scalable Sampling for Nonsymmetric Determinantal Point Processes"
_ICLR.cc/2022/Conference — ICLR 2022 Spotlight_

### Official Review · Reviewer_ccEk · 2021-10-21

**Correctness:** 4
**Technical Novelty And Significance:** 3
**Empirical Novelty And Significance:** 2
**Recommendation:** 8
**Confidence:** 4

**Main Review:**

The paper is well-written. Even though it is a somewhat straightforward mix of ideas from Gartrell et al. (motivating such NDPPS), Gillenwater et al. (sublinear sampling algorithm of symmetric DPPs), and Poulson (Cholesky-based sampling of DPPs and NDPPs); I believe that the results are interesting enough for acceptance. The main contribution of the paper is, to my eyes, to have found a symmetric DPP that is well-suited for rejection sampling. The proof that $\forall Y, det(L_Y)\leq det(\hat{L}_Y)$, essential for the rejection sampling framework to work, is an interesting contribution (and could be used in other works on DPPs as well). In passing, I find the proof hard to follow and suggest to the authors to do their best to find ways of simplifying it as much as possible for the camera-ready version. The simpler the proof, the more easily it could be transferred to other scenarios, increasing the potential impact of the paper.

As for Theorem 2, the orthogonality constraint is frustrating and I would imagine that further efforts could lift this constraint and obtain a bound that depends on how orthogonal both subspaces are (and recovering the current bound when they are indeed orthogonal). On the other hand, ONDPPs can indeed be motivated for learning as they indeed ensure that the full "available rank" is put to contribution. The empirical results tend to validate this intuitive argument.

**Summary Of The Paper:**

This paper gives an exact sampling algorithm to sample from DPPs based on non-symmetrical, low-rank, kernel matrices of the form $L=VV^\top + B(D-D^\top)B^\top$ where $V$ and $B$ are of size $M$ by $K$ and $D$ is square, non-symmetric and of size $K$ by $K$. The marginal kernel of such a DPP may be written in the form $K=ZWZ^\top$ with $Z$ a $M$ by $2K$ matrix and $W$ a square matrix of size $2K$ that is not symmetric. The authors then:
- discuss in Section 2 how Poulson's algorithm is adapted to sample from such NDPPS
- in Section 3: show that, as L is low-rank, the generic $O(M^3)$ cost of Poulson's algorithm can be reduced to $O(MK^2$ as updates of the marginal kernel can be efficiently done by only updating the inner $W$ matrix
- in Section 4: give a sublinear-time rejection-sampling based algorithm adapting the tree-based algorithm of Gillenwater et al. In this section, they:
	* propose a well-adapted, DPP based on a symmetric kernel $\hat{L}$ that is easy-to-sample (it is a DPP based on a symmetric PSD kernel and can be sampled from many different fast existing algorithms), and for which any set $Y$ verifies $det(L_Y)\leq det(\hat{L}_Y)$ where the upper-bound is actually attained (Thm 1). In order to exactly control the rejection rate of their proposed algorithm (that is equal to $det(\hat{L}+I) / det(L+I)$, the authors simplify the model by supposing that $V$ and $B$ are orthogonal to each other (Thm 2)
	* in passing, the authors observe that the complexity of Gillenwater et al's algorithm can in fact be easily improved by a factor $k$ where $k$ is the size of the sampled set
- in Sections 5 and 6, experiments are provided comparing their samping method with the state-of-the-art

**Summary Of The Review:**

All in all, the small lack of true originality is compensated by two useful theorems that can prove useful to the community.

---

> ### Author Response · Authors · 2021-11-17
> **Response to Reviewer ccEk**
>
> Thank you for the feedback and suggestions.  Regarding the points you raised:
> - The proof of Theorem 1 is hard to follow.
>
>   - We have updated our paper to make the proof of Theorem 1 simpler and easier to follow by highlighting its key ideas and adding more detailed descriptions of the steps.
>
>
> - The orthogonality constraint is frustrating and I would imagine that further efforts could lift this constraint and obtain a bound that depends on how orthogonal both subspaces are (and recovering the current bound when they are indeed orthogonal).
>
>   - We thank reviewer ccEk for this interesting idea, and agree that it is worth investigating. We will leave an exploration of this approach for future work.

---

> > ### Comment · Reviewer_ccEk · 2021-11-20
> > **ok**
> >
> > Dear authors,
> > good work with this paper. Thanks for the additional effort to simplify and better guide the reader through the proof of Thm 1.
> > Best,

---

### Official Review · Reviewer_uhQa · 2021-11-02

**Correctness:** 3
**Technical Novelty And Significance:** 3
**Empirical Novelty And Significance:** 3
**Recommendation:** 8
**Confidence:** 3

**Main Review:**

This paper proposes two scalable sampling algorithms for NDPPs, one for low-rank kernel, and the other for low rank orthogonal kernel. The efficiency of the proposed algorithms are verified through experiments, and the predictive performance is not degrade. This paper is well written, and the studies could benefit relevant researches on NDPPs.

I have some minor comments which should be easy to fix or answer:
- For Equ (4) and Equ (5), please give a definition of ${\bf z}_j$.
- In section 4.1, the eigendecomposition of ${\bf V}{\bf V}^T$ is not used in Algorithm 2. Could we remove it to avoid the confusion?
- In section 4.1, the dimensions of $\bf Z$ and $\bf X$ do not match when K is odd.
- In section 5, could you elaborate on why if ${\bf V}\not\perp{\bf B}$, $\bf L$ will have rank $<2K$? What happens if ${\bf V}={\bf V}_1+{\bf B}$ where ${\bf V}_1\perp{\bf B}$?


**Summary Of The Paper:**

This paper studies scalable sampling methods for NDPPs:
- By assuming low-rank structure of the kernel, the paper proposes a linear-time sampling method, which is faster than previous sampling algorithm with cubic runtime for general kernel.
- Furthermore, this paper develops a scalable sub-linear-time rejection sampling algorithm for a subclass of low-rank NDPPs that are called ONDPPs.
- Through experiments, it is shown that the predictive performance of ONDPPs is not degraded compared to NDPPs. By adding a regularization term in the optimization, the rejection probability is greatly reduced.

**Summary Of The Review:**

Overall I think this is a well written paper. My only concern is that the more scalable sampling algorithm is achieved by imposing more restrictions on the kernel, which might limit its usage. However, this might not be a big issue because it is shown in the experiments that its predictive performance is not degraded.

---

> ### Author Response · Authors · 2021-11-17
> **Response to Reviewer uhQa**
>
> Thank you for the feedback and suggestions.  Regarding the points you raised:
> - For Equ (4) and Equ (5), please give a definition of ${\boldsymbol z}_j$; and
> In section 4.1, the eigendecomposition of ${\boldsymbol V}{\boldsymbol V}^\top$  is not used in Algorithm 2. Could we remove it to avoid the confusion?
>
>   - Following your suggestion, we have updated our paper to resolve these issues.
>
>
> - In section 4.1, the dimensions of ${\boldsymbol Z}$ and $\boldsymbol X$ do not match when $K$ is odd.
>
>   - We note that nonsingular skew-symmetric matrices always have even rank ($K$), because all of the eigenvalues are purely imaginary and come in conjugate pairs. Therefore, it is not necessary to consider odd values of $K$.
>
>
> - In section 5, could you elaborate on why if ${\boldsymbol V}\not\perp {\boldsymbol B}$, ${\boldsymbol L}$ will have rank $<2K$?
>
>   - Actually, even if ${\boldsymbol V}\not\perp {\boldsymbol B}$, the rank of $\boldsymbol L$ can potentially be $2K$. A more precise condition for rank $< 2K$ is an existence of a column in ${\boldsymbol V}$ (or ${\boldsymbol B}$) spanned by the columns of ${\boldsymbol B}$ (or ${\boldsymbol V}$). If this holds, then the symmetric part of ${\boldsymbol L}$ can be expressed as a rank $< K$ matrix. We have updated our paper to correct this statement.
>
>
> - What happens if ${\boldsymbol V} = {\boldsymbol V}_1 + {\boldsymbol B}$ where ${\boldsymbol V}_1 \perp {\boldsymbol B}$?
>
>   - In this case, $\mathrm{rank}({\boldsymbol V}) = \mathrm{rank}({\boldsymbol V}_1) + \mathrm{rank}({\boldsymbol B})$, and since both ${\boldsymbol V}$ and ${\boldsymbol B}$ are $M$-by-$K$ matrices, ${\boldsymbol B}$ would be rank-deficient, which reduces the size of support subsets of NDPPs.
>
> - Concerned that the more scalable sampling algorithm requires restrictions on the kernel, which might limit its usage.
>
>   - Our sublinear-time rejection sampling algorithm can actually be applied to general low-rank NDPPs, as the restriction on the kernel is only for guaranteeing a small number of rejections. Also, as reviewer uhQa pointed out, our paper shows that enforcing this restriction does not degrade predictive performance.

---

> > ### Comment · Reviewer_uhQa · 2021-11-25
> > **Reply to response**
> >
> > - In section 4.1, the dimensions of $\bf Z$ and $\bf X$ do not match when $K$ is odd.
> >   - I agree that nonsingular skew-symmetric matrices always have even rank. But here the definition of $K$ is the rank of the symmetric part ${\bf V}{\bf V}^T$, which is not necessarily even. So the feedback here is that $\bf Z$ and $\bf X$ do not match when the rank of the symmetric part ${\bf V}{\bf V}^T$ and the rank of the skew-symmetric part are different.
> >
> > I am satisfied with all other responses. As the above is a minor comment which is easy to fix, I will increase my rating to 8.

---

### Official Review · Reviewer_MEEx · 2021-11-03

**Correctness:** 4
**Technical Novelty And Significance:** 3
**Empirical Novelty And Significance:** 3
**Recommendation:** 8
**Confidence:** 4

**Main Review:**

The paper is well written, clear, and interesting to read. They are the first to give efficient algorithms for exact sampling from NDPPs (the previous sampling algorithm takes time cubic in ground set size, which is impractical for real-world data). Their work can lead to more applications of NDPPs in practical settings. It is also timely and adds to the growing literature on NDPPs.

Some comments:
For equations 2 and 3 (and similar probabilities elsewhere in the paper which only involve singleton sets), it might help the reader if you use Pr$(i \in Y\ |\ j \in Y)$ rather than Pr$({i} \subseteq Y\ |\ \{j\} \subseteq Y)$. Also, for these specific equations, you mention that Poulson (2019) shows them via the Cholesky decomposition but that seems overkill. For these specific equations (2 and 3), you can just derive equation 2 by computing $\frac{Pr(i,j \subseteq Y)}{Pr(j \subseteq Y)} = \frac{\det (K_{i,j})}{\det(K_{j})}$ and equation 3 also by $\frac{Pr(i \in Y, j \notin Y)}{Pr(j \in Y)} = \frac{Pr(i \in Y) - Pr (i,j \subseteq Y)}{Pr(j \in Y)} = \frac{\det (K_{i}) - \det (K_{i,j})}{\det(K_{j})}$. It seems like Poulson (2019) uses LU decomposition (in Proposition 3 and 5) because their propositions are more general and apply to disjoint subsets (possibly having more than one element). Also, I'm not very familiar with these factorizations but seems like LU decomposition is different from Cholesky decomposition (at least from my understanding reading the wiki for Cholesky decomposition). Experts might be irritated by this (or maybe it's fine in which case, please let me know).

**Summary Of The Paper:**

The paper provides efficient (linear time in item set size) algorithms for exact sampling from Nonsymmetric Determinantal Point Processes (NDPPs) using a new NDPP decomposition which was introduced recently in an ICLR 2021 Oral (Gartrell et.al) and also a learning algorithm to learn NDPP kernels which are more amenable to some of their sampling algorithms. They also provide empirical results comparing their learned kernels with previous works and compare times for their various sampling algorithms.

**Summary Of The Review:**

Overall, I like the paper and recommend acceptance. This is the first paper to give efficient algorithms for exact sampling from NDPPs (which can be used in a variety of real-world applications like recommender systems etc) and thus this paper can have a good impact.

---

> ### Author Response · Authors · 2021-11-17
> **Response to Reviewer MEEx**
>
> Thank you for the feedback and suggestions.  Regarding the points you raised:
> - Write probabilities involving singleton sets with "$\in$" notation rather than "$\subseteq$". It might help the reader.
>   - Following your suggestion, we have updated the probability notation used for singleton sets with the “$\in$” notation.
>
>
> - LU decomposition is different from Cholesky decomposition
>   - As the reviewer pointed out, the Cholesky decomposition is indeed less general than the LU decomposition, since the Cholesky decomposition can only be applied to a symmetric positive definite matrix, and thus cannot be applied to NDPP nonsymmetric kernel matrices.  The LU decomposition is valid for any nonsingular matrix, and is therefore more appropriate for NDPPs.  We used the term “Cholesky” following Poulson (2019), but actually neither Poulson’s algorithm nor ours performs a Cholesky decomposition; the conditioning steps in the sampling algorithm are simply inspired by the steps required to do a Cholesky decomposition.  We have updated our paper to add a remark on this issue.
>
>     Poulson, Jack. "High-performance sampling of generic determinantal point processes." arXiv:1905.00165, 2019.
>
>
>
> - Thank you for discussing interesting derivations for Equations 2 and 3. We have updated our paper with these derivations.

---

> > ### Comment · Reviewer_MEEx · 2021-11-22
> > **Acknowledgement of author response**
> >
> > Hi, I just wanted to say that I have read the author response and the reviews by the other reviewers and my positive view of the paper hasn't changed. It's also good that the authors have improved the paper by taking the reviews into account.

---

### Official Review · Reviewer_Tzr3 · 2021-11-04

**Correctness:** 4
**Technical Novelty And Significance:** 3
**Empirical Novelty And Significance:** 2
**Recommendation:** 8
**Confidence:** 3

**Main Review:**

This paper is overall well-written, easy to understand and both the theoretical and empirical results presented in this paper are very exciting.

Compared to the linear-time Cholesky-decomposition-based sampler, the rejection-sampling-based method is definitely the main focus of the paper. Though the rejection-sampling method is not strictly sub-linear-time in general, the theoretical result presented in this paper (Theorem 2) directly implies that in practice, we can easily allow efficient sampling by bounding the expected # of rejections in learning NDPPs (ONDPPs); more importantly, the authors conducted comprehensive experiments to show that in learning NDPPs, by imposing the constraints (orthogonality & extra regularization term) we need for efficient sampling, we are not losing the expressive power of NDPPs. Experiments also show that compared to the linear-time sampling algorithm, the rejection-sampling method scales much better on both synthetic and real-world datasets.

NDPPs are a strictly more expressive class compared to DPPs but the absence of efficient sampling algorithms puts a major barrier from replacing DPPs by NDPPs in large-scale applications. By proposing an efficient "sub-linear-time" sampling algorithm for NDPPs, this paper makes substantial progress in scaling up NDPPs and opens up new avenues for applying NDPPs to various real-world scenarios.

Cons:
1. Compared to the rejection-sampling method, the linear-time algorithm is not as interesting and novel and somewhat deviating from the main story. Authors might want to consider presenting the linear-time algorithm without getting into the technical details. Of course, the experiments that compares the rejection-sampling method against the linear-time algorithm is still important.

2. Section 4.2 is a little bit difficult to follow for those who are not familiar with the tree-based sampling algorithm for DPPs. Authors might want to expand this section a little bit.


**Summary Of The Paper:**

This paper studies the problem of sampling from non-symmetrical Determinantal Point Processes (NDPPs); in particular, this paper focuses on exact sampling for low-rank NDPPs.

The main contributions of this paper are:
1. This paper proposes to adapt the Cholesky-decomposition-based sampler for DPPs to a linear-time sampler for low-rank NDPPs (rank of NDPP << size of the ground set)
2. This paper proposes to use rejection sampling to implement a "sub-linear-time" (assuming that the # of rejections are bounded by a small constant) sampler for a subclass of NDPPs called orthogonal NDPPs (ONDPPs), by leveraging an existing sub-linear-time sampler for DPPs.
3. This paper shows that empirically, in terms of modeling real-world datasets, ONDPPs are as effective as the general NDPPs and more importantly, we can efficiently learn ONDPPs in a way such that when it comes to rejection sampling, the # of rejections is bounded by a small number.

**Summary Of The Review:**

By proposing an efficient (sub-linear-time under certain assumptions) exact sampling algorithm for NDPPs, this work makes substantial progress in scaling up NDPPs to real-world applications. This paper is overall well-written and easy to follow, and it also demonstrates a decent amount theoretical novelties.

---

> ### Author Response · Authors · 2021-11-17
> **Response to Reviewer Tzr3**
>
> Thank you for the feedback and suggestions.  Regarding the points you raised:
> - Authors might want to consider presenting the linear-time algorithm without getting into the technical details.
>   - We agree that our main contribution is the sublinear-time rejection sampling algorithm.  However, the linear-time sampling has never been studied and requires a smaller memory space than the rejection sampling approach.  We believe that it is useful for settings where reduced memory consumption is important.
>
>
> - Section 4.2 is a little bit difficult to follow for those who are not familiar with the tree-based sampling algorithm for DPPs. Authors might want to expand this section a little bit.
>   - This is a good point.  We have refined the presentation of the tree-based sampling algorithm in section 4.2 so that it is easier to read and understand for those with no background on this algorithm.

---

> > ### Comment · Reviewer_Tzr3 · 2021-11-29
> > **Reply to response**
> >
> > My positive rating for this paper hasn't changed. I thank authors for addressing the issues with presentations and making this paper even better.

---

### Decision · Program_Chairs · 2022-01-20

**Decision:**

Accept (Spotlight)

**Comment:**

This is an exciting paper that provide the efficient algorithms for exact sampling from NDPPs along with theoretical results that are very pertinent in and out themselves. The AC agree with the reviewers that the authors satisfactorily addressed the concerns raised in the reviews, and is convinced that the revised version will be greatly appreciated by the community. We very much encourage the authors to pursue this line of work and in particular to overcome the practical restriction to the ONDPP subclass.